# Surface energy fluxes on Chilean glaciers: measurements and models

Marius Schaefer[1], Duilio Fonseca-Gallardo[1], David Farías-Barahona[2], and Gino Casassa[3,4]

[1]Instituto de Ciencias Física y Matemáticas, Facultad de Ciencias, Austral University, Valdivia, Chile
[2]Institut für Geographie, Friedrich-Alexander-Universität Erlangen-Nürnberg, Erlangen,Germany
[3]Dirección General de Aguas, Ministerio de Obras Públicas, Santiago, Chile
[4]Universidad de Magallanes, Punta Arenas, Chile

**Correspondence:** Marius Schaefer (mschaefer@uach.cl)

**Abstract.** The surface energy fluxes of glaciers determine surface melt and their adequate parametrization is one of the keys for a successful prediction of future glacier mass balance and freshwater discharge. Chile hosts glaciers in a large range of latitudes under contrasting climatic settings: from 18°S in the Atacama Desert to 55°S on Tierra del Fuego Island, Southern Patagonia. Using three different methods, we computed surface energy fluxes for five glaciers which represent the main glaciological

zones of Chile. We found that the main energy sources for surface melt change from the Central Andes, where the net shortwave radiation is driving the melt, to Patagonia, where the turbulent fluxes are an important source of energy. We inferred higher surface melt rates for Patagonian glaciers as compared to the glaciers of the Central Andes due to a higher contribution of the turbulent sensible heat flux, less negative net longwave radiation and a positive contribution of the turbulent latent heat flux. The variability of the atmospheric emissivity was high and not able to be explained exclusively by the variability of the inferred

cloud cover. The influence of the stability correction and the roughness length on the magnitude of the turbulent fluxes in the different climate settings was examined. We conclude that, when working towards physical melt models, it is not sufficient to use the observed melt as a measure of model performance: the model parametrizations of individual components of the energy balance have to be validated individually against measurements.

**1   Introduction**

Glaciers are retreating and thinning in nearly all parts of the planet and it is expected that these processes are going to continue under the projections of global warming (IPCC, 2019). For mountain glaciers melt is mostly determined by the energy exchange with the atmosphere at its surface. The processes leading to this exchange of energy are complex and depend on the detailed (micro-) climate on the glacier. Classical empirical melt models like for example degree day models (Braithwaite, 1995a) are

getting more and more replaced by more complex models which try to quantify the detailed physical processes that govern

the energy exchange at the glacier surface. These kind of models are sometimes called "physical melt models" or "physically based models" (Pellicciotti et al., 2008).

In Chile, the only glacier with a climatologically relevant long-term record of surface mass balance is Echaurren Norte Glacier near to Santiago de Chile, which is monitored since 1975 (WGMS, 2017; Masiokas et al., 2016; Farías-Barahona et al., 2019). Echaurren Norte Glacier (33.5°S) has a general negative trend in its cumulative surface mass balance (-0.48 m w.eq./year in 1976-2017) but also shows stable phases in the 1980s and the first decade of the 21st century, (Masiokas et al., 2016; WGMS, 2017; Farías-Barahona et al., 2019). The variations of the surface mass balance of this glacier can be mostly explained by variations of precipitation in the region (Masiokas et al., 2016; Farías-Barahona et al., 2019). In the semi arid Pascua Lama Region (29°S) several small glaciers have been monitored since 2003 (Rabatel et al., 2011). These glaciers also show mostly negative surface mass balance and are losing area (Rabatel et al., 2011). During the monitoring period, the limited accumulation of snow is not able to make up with the ablation which is dominated by sublimation (MacDonell et al., 2013). In the Chilean Lake District, Mocho Glacier is monitored since 2003 (Rivera et al., 2005). Here, a very high inter-annual variability of the surface mass balance was observed (Schaefer et al., 2017). But, on average, the annual surface mass balance was negative which coincides with the observed areal losses (Rivera et al., 2005).

Energy balance studies have been realized in Chile on different glaciers: in the semiarid Andes MacDonell et al. (2013) quantified in detail the drivers of ablation processes on Guanaco Glacier (29°S). They found that the net shortwave radiation is the main source and that the net longwave radiation and turbulent flux of latent heat are the main sinks of energy at the surface of Guanaco Glacier (MacDonell et al., 2013). Due to the low temperatures at this high elevation site (5324 m a.s.l), they found that sublimation dominated the surface ablation and surface melt contributed only during summer. Pellicciotti et al. (2008) and Ayala et al. (2017b) studied the surface energy balance during summer at Juncal Norte Glacier in the Central Andes (33°S, near Santiago de Chile. Similar to MacDonell et al. (2013) they found that the net shortwave radiation is the main source and that the net longwave radiation and turbulent flux of latent heat are the main sinks of energy. Similar results concerning the influence of the different components of the surface energy balance where obtained by Ayala et al. (2017a), who analyzed meteorological data collected on six glaciers in the semiarid Andes of North-Central Chile at elevations spanning from 3127 m.a.s.l. to 5324 m.a.s.l..

Brock et al. (2007) studied the surface energy balance of bare snow and tephra-covered ice on Pichillancahue-Turbio Glacier (39.5°S) on Villarrica Volcano in the Chilean Lake District during two summers. They found a strong reduction of surface melt on the tephra-covered part of the glacier and a change in sign of the turbulent flux of latent energy to a source due to the higher vapor pressure caused by a more humid atmosphere as compared to the northern and central part of Chile. In southernmost Chile, Schneider et al. (2007) studied the energy balance in the ablation area of Lengua Glacier, which is an outlet glacier of Gran Campo Nevado Ice Cap (53°S). They found that during February to April 2000, due to the high air temperatures and the high wind speeds turbulent flux of sensible heat was the main source of melt energy for the glacier surface.

In a comparative study of the surface energy balance of glaciers at different latitudes, Sicart et al. (2008) found that the net shortwave radiation is driving the glacier melt at the tropical Zongo Glacier, but that at Storglaciären in Northern Sweden the turbulent fluxes of sensible heat and latent heat dominated the melt patterns.

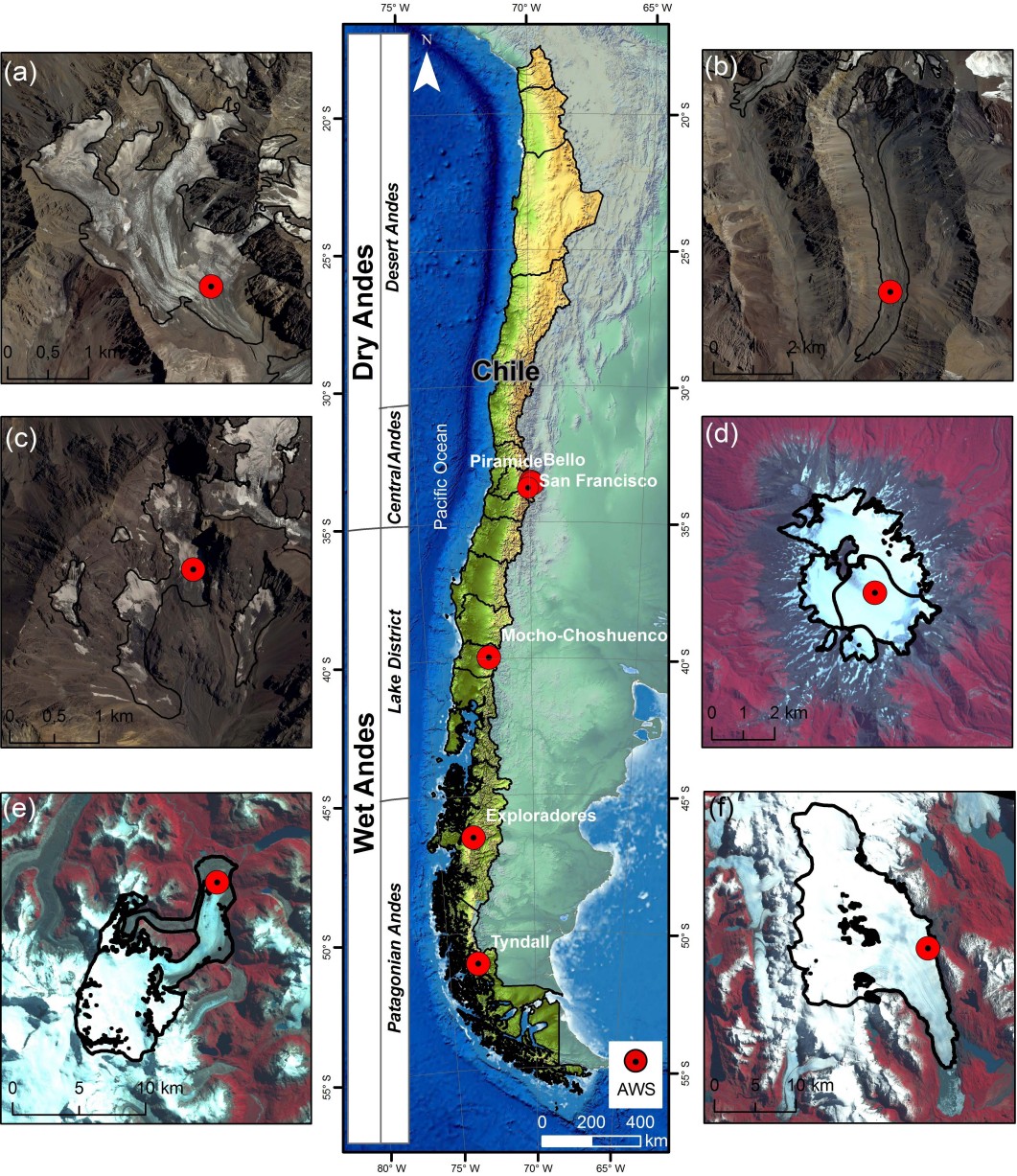

**Figure 1.** Middle: glaciological zones in Chile according to Lliboutry (1998) and locations of the studied glaciers: (a) Bello Glacier, (b) Pirámide Glacier, (c) San Francisco Glacier, (d) Mocho Glacier, (e) Exploradores Glacier, (f) Tyndall Glacier.

In this study we analyze data from automatic weather stations (AWSs) installed on six glaciers distributed in the different glaciological zones of Chile (Figure 1), five of them being equipped and maintained by the Unit of Glaciology and Snows of the Chilean Water Directory (UGN-DGA) (UChile, 2012; Geoestudios, 2013; CEAZA, 2015). Using the meteorological observations as input, we compare different ways to compute the glacier surface energy balance: we use direct measurements of the radiative fluxes at the glacier surface and two models that are freely available: the spreadsheet-based point surface energy balance model (EB-model) developed by Brock and Arnold (2000) and the Coupled Snowpack and Ice surface energy Mass balance model (COSIMA) (Huintjes et al., 2015c, a).

Instead of validating the ability of the energy balance calculations to adequately predict melt rates, in this study we want to test their ability to reproduce the individual energy fluxes: we want to emphasize the differences between the model parametrizations and their ability to reproduce the directly measured radiative fluxes at the glacier surfaces. We also compare different parametrizations for the turbulent fluxes of sensible and latent heat and discuss the influence of stability corrections and roughness lengths.

## 2 Sites

The projections of future changes in climate depend on the different climatological/glaciological zones. This is why a detailed analysis of the processes that determine the energy exchange at the surface of the glaciers in the different climatological zones is necessary, to be able to make reliable predictions of future surface mass balance and melt water discharge of Chilean glaciers.

Chile's climate is strongly determined by the Pacific Anticyclone and the Andes Range which acts as a natural barrier (Fuenzalida-Ponce, 1971; Garreaud, 2009). Large climate differences are observed due to the large north-south extent of the territory (4000 km, 17°30 – 55°S). Despite the different classifications of sub-glaciological zones (Lliboutry, 1998; Masiokas et al., 2009; Barcaza et al., 2017; Braun et al., 2019; Dussaillant et al., 2019) most authors agree that there is a transition from Dry Andes to Wet Andes at around 35°S (Figure 1.). The Central Andes of Chile (31°-35°S) are characterized by a Mediterranean climate, with dry conditions during summer. For the period 1979–2006 Falvey and Garreaud (2009) observed a cooling in the coast and a considerable temperature increase of +0.25°C/decade inland in the Maipo River catchment in the Central Andes. Precipitation in this area is highly variable, and predominantly occurs during winter (Falvey and Garreaud, 2007) controlled by El Niño Southern oscillation (ENSO) and the Southeast Pacific Anticyclone (Montecinos and Aceituno, 2003). Between 2010 and 2015 a mega-drought was observed in the Central Andes (Boisier et al., 2016; Garreaud et al., 2017).

In the northern part of the Wet Andes (35°-45°S), known in Chile as the Lake District, the elevation range steadily decreases, and wetter climatic conditions are predominant. A general decrease of precipitation in the region was observed during the 20th century (Bown et al., 2007; González-Reyes and Muñoz, 2013). The Southern part of the Wet Andes, Patagonia, is characterized by a hyper-humid climate (Garreaud, 2018), where the largest glacierized areas in the Southern Hemisphere outside Antarctica can be found. This hyper-humid condition has been recently interrupted by a severe drought during 2016 with a precipitation decrease of more than 50% (Garreaud, 2018). Under these different climatic settings, Chile hosts the majority of glaciers in

South America (more than 80% of the area), which are mostly thinning and retreating in the last decades (e.g. Braun et al. (2019); Dussaillant et al. (2019)).

In the Central Andes, San Francisco ($1.5\,km^2$) and Bello ($4.2\,km^2$) Glaciers are mountain glaciers which are partially debris-covered at their termini and Pirámide Glacier ($4.4\,km^2$) is an almost completely debris-covered glacier (Figure 1). On San Francisco and Bello Glacier the AWSs were installed over bare ice and at Pirámide Glacier they were installed over debris-cover. Mocho Glacier is part of the ice cap ($14\,km^2$) which is covering the Mocho-Choshuenco volcanic complex, located in the Lake District (Schaefer et al., 2017). Exploradores Glacier ($83.8\,km^2$) is located on the northern margin of the Northern Patagonia Icefield with a prominent portion of debris-cover at its tongue. Recently, at the glacier's front, several lateral lakes have developed and some calving activity was observed. Finally, Tyndall Glacier ($309.8\,km^2$) is one of the large glaciers in the Southeastern part of the Southern Patagonia Icefield. Tyndall Glacier is terminating in Geike Lake, where it experiences additional mass losses through calving. All glacier areas are from Barcaza et al. (2017).

Due to the installation of AWSs on several glaciers in the country by the UGN-DGA, detailed meteorological observations from glaciers in the different glaciological zones are available now (UChile, 2012; Geoestudios, 2013; CEAZA, 2015). In Table 1 we present the detailed locations of the AWSs used for this study and some relevant glacier parameters.

**Table 1.** Study period and geographical information of the glaciers and automatic weather stations.

| Glacier | Period | Latitude | Longitude | Elevation | ELA | Exposure |
|---------|--------|----------|-----------|-----------|-----|----------|
| Name | | ° | ° | m a.s.l. | m a.s.l. | |
| Bello | 01/01/2015-31/03/2015 | -33.53 | -69.94 | 4134 | 4600[Ayala et al., 2016] | SE |
| Pirámide | 01/01/2016-31/03/2016 | -33.59 | -69.89 | 3459 | 3970[Ayala et al., 2016] | S |
| San Francisco | 01/01/2016-31/03/2016 | -33.75 | -70.07 | 3466 | 3970[Carrasco et al., 2008] | SE |
| Mocho | 31/01/2006-21/03/2006 | -39.94 | -72.02 | 2003 | 1990[Schaefer et al., 2017] | SE |
| Exploradores | 01/01/2015-31/03/2015 | -46.51 | -73.18 | 191 | 1420[Schaefer et al., 2013] | N |
| Tyndall | 01/01/2015-31/03/2015 01/01/2016-31/03/2016 | -51.13 | -73.31 | 608 | 1020[Schaefer et al., 2015] | SE |

Because of its higher relevance for melt modeling, we focused our analysis to summer periods: AWSs of the UGN-DGA have a data record of several years, but during several summers some of the sensors were not working well. In Table 1 we show the selected summer period for every station. For Tyndall Glacier two summers were analyzed. On Mocho Glacier an AWS was installed only during a 50 day period during summer 2006. Figure 2 shows photos from the AWSs installed on the glaciers Bello, Exploradores and Tyndall.

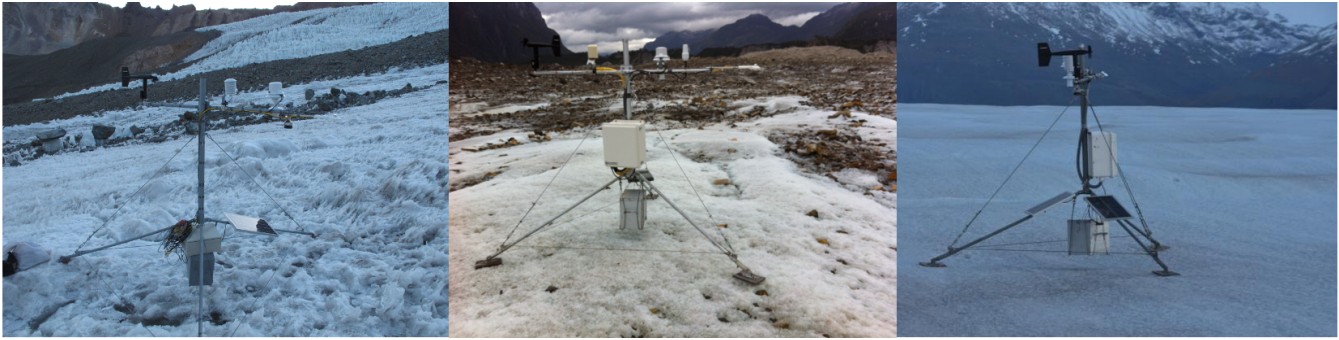

**Figure 2.** AWS on Bello Glacier (left), Exploradores Glacier (middle) and Tyndall Glacier (right)

## 3   Methods

In this contribution we want to focus on the six most important energy fluxes which normally determine the melt energy available at the glacier surface: the incoming solar radiation ($SW_{in}$), the reflected solar radiation ($SW_{out}$), the incoming atmospheric longwave radiation ($LW_{in}$), the longwave radiation emitted by the glacier surface ($LW_{out}$) and turbulent fluxes of sensible ($SH$) and latent heat ($LH$). We will compare three methods to compute these surface energy fluxes for the selected sites. Before presenting the three methods in detail, we describe the input data for our energy balance calculation in the following section.

### 3.1   Input data

In Table 2 we present the sensors used at the Mocho-AWS in 2006 and the instruments used at the DGA stations on the other glaciers. The main difference in the installation is the CNR4 sensor, which is installed on the DGA stations and provides detailed measurements of all radiative fluxes, while on Mocho Glacier $SW_{in}$, $SW_{out}$ and the net allwave radiation is measured. At Mocho-AWS mean values of the data were recorded every 15 minutes, which were resampled to hourly data for the energy balance calculations. At the DGA stations hourly means are recorded and transmitted by a satellite connection. Data at missing hours were interpolated by taking the mean value of the hour before and after the missing one. For Bello Glacier the time resolution of the acquired data changed from hourly to three hourly on 20th of March 2015. Hourly data were generated using the linear interp1 matlab interpolation scheme.

### 3.2   Reference Database

We call this first method the Reference Database, since in this approach direct measurements of the first four fluxes (the radiative fluxes) are used (with the exception of the Mocho-AWS where the net longwave radiative flux is inferred from the incoming solar radiation, the reflected solar radiation and the overall net radiative flux). However, on several glaciers the measured mean outgoing longwave radiative fluxes were higher then the ones expected for a blackbody at zero degrees Celsius. Therefore we decided to bias-correct the measured longwave radiative fluxes in a way that the measured outgoing longwave radiative fluxes in the afternoon correspond to a melting surface at zero degrees Celsius. From this calibration of the signal we obtained

**Table 2.** Sensors employed at the different AWSs

| Variable | AWS-DGA | nominal accuracy | AWS Mocho | nominal accuracy |
|---|---|---|---|---|
| Incoming solar $SW_{in}$ | Kipp & Zonen CNR 4 | 7-8% on daily total | Kipp & Zonen SP-Lite | 7-8% on daily total |
| Reflected solar $SW_{out}$ | Kipp & Zonen CNR 4 | 7-8% on daily total | Kipp & Zonen SP-Lite | 7-8% on daily total |
| Incoming longwave $LW_{in}$ | Kipp & Zonen CNR 4 | 7-8% on daily total | - | - |
| Outgoing longwave $LW_{out}$ | Kipp & Zonen CNR 4 | 7-8% on daily total | - | - |
| Net all wave $AW_{net}$ | - | - | NR- lite | $\pm 3\%$ |
| Air Temperature $T$ | HMP60 | $\pm 0.6^\circ C$ | HMP45c | $\pm 0.3^\circ$C |
| Relative Humidity $RH$ | HMP60 | max 7% | HMP45c | max 3% |
| Wind Speed $U$ | Young 05103 | 0.3 m/s o $\pm 1\%$ | Young 05103 | $\pm$ 0.3 m/s o $\pm 1\%$ |

different correction factors for the longwave radiative fluxes which were applied to both incoming and outgoing longwave radiative fluxes (Table 4).

The turbulent fluxes of sensible and latent heat were calculated according to formulas derived in Cuffey and Paterson (2010). The bulk aerodynamic approach is employed and the following important assumptions are made.

5     1. The eddy diffusivity for heat has the same value as the eddy diffusivity for water vapor and the eddy viscosity.

    2. The shear stress in the first few meters of the atmosphere above the glacier surface is constant.

    3. The wind velocity, temperature and water vapor pressure have logarithmic profiles with the same scaling length $z_0$.

Using these assumptions, the following expression for the turbulent flux of sensible heat can be derived:

$$SH = c_a \rho_a C^*(z) U(z) \left[ T(z) - T(s) \right], \tag{1}$$

10   where $c_a$ is the specific heat of air at constant pressure which was assumed to be constant at 1.01 kJ/(kg K) , $\rho_a$ is the air density, $U(z)$ is the wind speed measured at the height $z$ above the surface, $T(z)$ is the air temperature at the height $z$ of the sensor and $T(s)$ is the temperature of the glacier-atmosphere interface.

The dimensionless number $C^*(z)$ is a proportionality constant called the transfer coefficient. If the above assumptions are fullfilled, it should depend on the measurement height of the sensors of wind velocity and temperature $z$ (two meters in our 15   case) and the roughness length $z_0$ according to:

$$C^*(z) = \frac{\kappa^2}{ln^2(z/z_0)}, \tag{2}$$

where $\kappa$ is the von Karman constant, which has an approximate value of 0.4. In practice however the roughness/scaling length $z_0$ is variable in space and time (Brock et al., 2006). There exist several recommendations in the literature of values for $C^*(z)$ that have produced satisfying results (Cuffey and Paterson, 2010), which gives $C^*(z)$ rather the interpretation of a tuning 20   parameter than a physical constant. In the results section we present the results obtained by using an intermediate roughness

**Table 3.** Variation of the transfer coefficient $C^*(2\,\text{m})$ for typical values of the roughness length $z_0$

| Roughness length in mm | $C^*(2m)$ |
|:---:|:---:|
| 0.01 | 0.001 |
| 0.5 | 0.0023 |
| 1 | 0.0028 |
| 5 | 0.0045 |
| 10 | 0.0057 |
| 30 | 0.009 |

length of $z_0$=0.5 mm for all the glaciers. According to table 5.4 in Cuffey and Paterson (2010) this corresponds to a value between smooth ice and ice in the ablation zone and is also inside the range recommended for new and polar snow. When looking at the glacier surfaces in Figure 2, we can note that the roughness length is probably varying from glacier to glacier and in Table 3 we present how $C^*(2\,\text{m})$ should vary for typical values of $z_0$.

Assumption 1. is normally fulfilled for a neutral atmosphere, an atmosphere which exhibits a temperature lapse rate equal to the dry adiabatic lapse rate of 9.8 °C/km. Over a glacier surface, the temperature gradient is often inverted (temperature increases with elevation), especially during summer when the air temperature is positive. This stable layering of air masses reduces the vertical exchange specially for low wind speeds and we apply a stability correction to equation (1) which depends on the bulk Richardson number $Ri = \frac{gT(2m)2m}{(T(2m)+273.15)U^2}$, where $g$ is gravitational acceleration and $T(2m)$ has to be taken in

degrees Celsius. The correction factor is smaller than one for $Ri > 0.01$ (small wind speeds) and approaches zero for $Ri = 0.2$ in the same way as it is implemented in the COSIMA model (Huintjes et al., 2015a).

Using the same arguments from above and assuming the turbulent flux of latent heat to be proportional to the difference of the concentration of water vapor at the glacier surface and the air layer above it,Cuffey and Paterson (2010) derive the following expression :

$$LH = 0.622\rho_a L_v C^* U(z)\left[P_{\text{vap}}(z) - P_{\text{vap}}(s)\right]/P_a. \tag{3}$$

Here, $P_{\text{vap}}(z)$ and $P_{\text{vap}}(s)$ are the water vapor pressure at the elevation $z$=2 m above the glacier and at its surface respectively, $P_a$ is the air pressure and $L_v$ is the latent heat of vaporization. The water vapor pressure at $z$=2 m depends on the (measured) relative humidity and the saturation water vapor pressure $P_{\text{vap,sat}}$ which depends on the air temperature. In Figure 3.(a) we show measurements of the saturation water vapor pressure at different temperatures (Lide, 2004) and the graphs of several

parametrizations as a function of the air temperature, found in the literature (Bolton, 1980; Cuffey and Paterson, 2010; Huintjes et al., 2015a). We decided to use the parametrization proposed in Bolton (1980), since it agrees best with the measurements:

$$P_{\text{vap,sat}}(T) = 6.112 \exp\left(\frac{17.67\,T}{T + 243.5}\right), \tag{4}$$

where $P_{\text{vap,sat}}$ is in hectopascal and the air temperature $T$ is in degrees Celsius. It is assumed that at the glacier surface the water vapor pressure is equal to the saturation vapor pressure at the surface temperature $T(s)$. The turbulent flux of latent heat is

corrected in the same way for stability conditions found over glacier surfaces as the turbulent flux of sensible heat (see above).

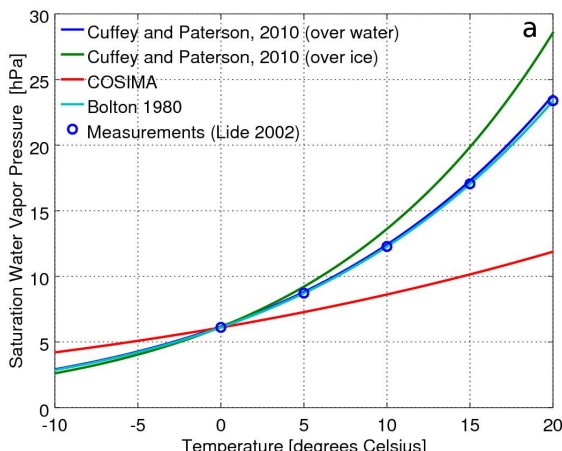 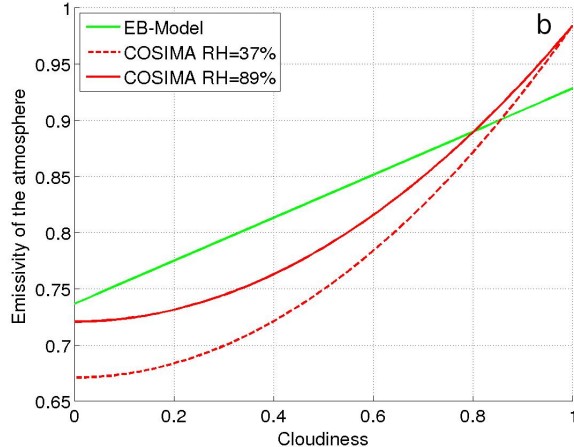

**Figure 3.** (a) Different parametrizations of the saturation vapor pressure as a function of the air temperature found in the literature; we chose the parametrization of Bolton (1980), (b) emissivity of the atmosphere as a function of the cloud cover as implemented in the EB-Model and COSIMA.

### 3.3 EB-Model

In the spreadsheet-based energy balance model developed by Brock and Arnold (2000) the incoming solar radiation, two meter air temperature, the wind speed and the water vapor pressure are the meteorological input variables. The fixed input parameters are latitude, longitude and elevation of the station, the aspect and slope, the albedo $\alpha$ and the roughness of the surface $z_0$.

5 The net shortwave radiation is calculated by multiplying the sum of the direct and diffuse incoming solar radiation by 1-$\alpha$. The incoming direct and diffuse incoming solar radiation depend on the measured incoming solar radiation $SW_{\text{in}}$ and the glacier's surface slope and aspect at the AWS. However, in our study, these parametrizations produced erroneous values for the net shortwave radiation $SW_{\text{net}}$ in the late afternoon for several glaciers. This is why we computed $SW_{\text{net}}$ from the measured incoming solar radiation $SW_{\text{in}}$ by multiplying it with 1-$\alpha$:

$$SW_{\text{net}} = (1 - \alpha)SW_{\text{in}}, \tag{5}$$

where the albedo $\alpha$ is assumed to be a constant, which depends on the characteristics of the glacier surface. This parametrization should exactly agree to the parametrizations proposed in Brock and Arnold (2000) for flat surfaces (zero slope), which should be a very good approximation, since the AWSs are normally placed on flat terrain.

The net longwave radiation is computed by assuming that the snow/ice surface irradiates thermal radiation of a black body

15 at 273.15 Kelvin (0 degrees Celsius) which is 315.6 W/m$^2$ according to the Stephan-Boltzmann law of thermal radiation. This value is subtracted from the incoming longwave radiation from the atmosphere which is computed with the Stefan-Boltzmann law as well, using the $T(2m)$ and an atmospheric emissivity which is a function of the cloud cover. Brock and Arnold (2000)

use a parametrization of the atmospheric emissivity $\varepsilon$ which increases linearly as a function of the cloudiness $n$:

$$\varepsilon_{(EB)}(n,T) = (1+0.26n)\varepsilon_{cs}(T), \tag{6}$$

where $\varepsilon_{cs}(T)$ is the clear sky emissivity which depends on the air temperature $T$: $\varepsilon_{cs}(T) = 0.00877T^{0.788}$ ($T$ in Kelvin). The green line in Figure 3 (b) shows the graph of $\varepsilon_{(EB)}(n,T)$ at five degree Celsius. The cloudiness is inferred by comparing the theoretically site-specific clear sky incoming solar radiation with the measured incoming solar radiation.

Similar to the Reference Database, in the EB-Model the turbulent fluxes of latent and sensible heat are calculated by expressions derived from the bulk aerodynamic method (Brock and Arnold, 2000) according to the equations (1) and (3). However, in this model the transport coefficient for the sensible heat flux $C^*_{EB1}$ and latent heat flux $C^*_{EB2}$ have a more complex form and do not only depend on the roughness length $z_0$ but also on the Monin-Obukhov length scale $L$ and the scaling lengths of temperature $z_T$ and humidity respectively $z_H$(Brock and Arnold, 2000):

$$C^*_{EB1} = \frac{\kappa^2}{(ln(z/z_0)+5z/L)(ln(z/z_T)+5z/L)} \tag{7}$$

$$C^*_{EB2} = \frac{\kappa^2}{(ln(z/z_0)+5z/L)(ln(z/z_H)+5z/L)} \tag{8}$$

$z_H$ and $z_T$ are calculated as a function of $z_0$ and the roughness Reynolds number (Brock and Arnold, 2000; Andreas, 1987). Since normally $z_H < z_T < z_0$ (Brock, 2018) and $L > 0$, $C^*_{EB1}$ and $C^*_{EB2}$ are smaller than $C^*$ and $C^*_{EB2} < C^*_{EB1}$.

## 3.4   COSIMA

The COupled Snow and Ice Melt MAss balance model (COSIMA) was developed at RWTH Aachen (Huintjes et al., 2015c, a) and combines a surface energy balance model with a multi-layer subsurface snow and ice model to compute glacier mass balance (Huintjes et al., 2015a). In this work we want to focus on how COSIMA models the six dominant energy fluxes at the glacier surface. The input parameters for the COSIMA model are the incoming solar radiation ($SW_{in}$), the two meter air temperature ($T$), the relative humidity ($RH$), the wind speed ($U$), the solid precipitation( $P_s$), the initial snow height, the air pressure ($P$) and the cloud cover ($n$) (Huintjes et al., 2015a). The daily mean cloud cover over the glacier was estimated by comparing the measured $SW_{in}$ with the theoretical, site specific clearsky radiation computed by a code developed by Corripio (2003). The cloud cover was determined from this cloud transmissivity $\tau_{cl}$ by solving the equation proposed in Greuell et al. (1997):

$$\tau_{cl} = 1 - 0.233n - 0.415n^2. \tag{9}$$

The net solar radiation is calculated using equation 5. In contrast to the EB-Model the albedo is variable and depends on the time since the last snowfall $t_{snow}$ and the thickness $h$ of the snow or firn layer on top of the glacier ice:

$$\alpha = \alpha_{snow} + (\alpha_{ice} - \alpha_{snow})\exp(-h/d^*), \tag{10}$$

where $\alpha_{\text{ice}}$ and $d^*$ are constants and

$$\alpha_{\text{snow}} = \alpha_{\text{firn}} + (\alpha_{\text{frsnow}} - \alpha_{\text{firn}}) \exp(t_{\text{snow}}/t^*), \tag{11}$$

with $\alpha_{\text{firn}}$, $\alpha_{\text{frsnow}}$ and $t^*$ being constants as well. This parametrization of snow albedo in COSIMA was tested at Mocho Glacier, where the glacier surface was covered by snow or firn during the observation period and precipitation data from a near-by
automatic weather station were available. On the other glaciers an ice surface and a constant albedo of 0.3 was assumed.

The longwave radiative fluxes are computed using the Stefan-Boltzmann law of thermal radiation as well. The snow/ice surface is considered as a blackbody. However, in contrast to the EB-Model in COSIMA the snow/ice surface temperature is variable depending on the heat fluxes at the glacier-atmosphere interface. Similar to the EB-Model the emissivity of the atmosphere is modeled as a function of the cloudiness($n$) using the following expression:

$$\varepsilon_{(\text{COS})}(n, T, P_{vap}) = \varepsilon_{cs}(T, P_{vap})(1 - n^2) + 0.984n^2, \tag{12}$$

where the emissivity of the clear sky depends on the air temperature and the water vapor pressure according to the following expression $\varepsilon_{cs}(T, P_{vap}) = 0.23 + 0.433(P_{vap}/T)^{1/8}$, where $T$ is in kelvin and $P_{vap}$ in pascal. The red graph in Figure 3 (b) shows the variation of $\varepsilon_{(\text{COS})}$ as a function of the cloud cover at five degrees Celsius and assuming different relative humidities.

The turbulent flux of sensible heat in COSIMA $SH_{(\text{COS})}$ is calculated using formula (1) using the modeled surface temper-
ature on the glacier surface. The same stability correction based on the bulk Richardson number $Ri$ describe in section 3.2 is applied here to account for the reduced vertical exchange of air masses in stable conditions (Braithwaite, 1995b).

The turbulent flux of latent heat is calculated by the following expression:

$$LH_{(\text{COS})} = 0.622\rho_a L_v C^* U(z) \left[ \frac{P_{\text{vap}}(z)}{P_a - P_{\text{vap,sat}}(z)} - \frac{P_{\text{vap}}(s)}{P_a - P_{\text{vap,sat}}(s)} \right]. \tag{13}$$

Since the air pressure $P_a$ is normally much higher than $P_{\text{vap,sat}}$, this formula should give very similar results as formula (3).
This expression is multiplied by the same correction factor as $SH$ (see section 3.2). Concerning the parametrization of $P_{\text{vap,sat}}$ as a function of temperature, we decided to replace the original parametrization in COSIMA (red line in Figure 3(a)) by the parametrization proposed by Bolton (1980), equation (4), since it agrees best with the measurements. The original parametrization of COSIMA underestimates the water vapor pressure at positive temperatures and therefore underestimates $LH$ for positive air temperatures, which are measured during summer at the AWSs (see below). COSIMA is modeling additional energy fluxes
like heat fluxes inside the snow or ice, but for a better comparison of the computed melt rates by the different methods, these fluxes were not considered in this contribution. The modelled heat flux inside the snow/ice with COSIMA depended on the initial temperature distribution inside the snow/ice and was maximum at San Francisco Glacier where it was 3% of the sum of the modelled fluxes, which are considered in this study.

## 4 Results

### 4.1 Microclimatic conditions on the glacier surface

In Table 4 we show averages of relevant climatic and glacier surface properties during summer for the six studied glaciers which are ordered according to their latitude from North to South. Variability of conditions is observed between the different

**Table 4.** Mean values of relevant meteorological and glacier surface data during the study periods. For the longwave radiative fluxes bias-corrected value is indicated as well as the original measured one in parenthesis.

| **Glacier** | $SW_{in}$ | $\alpha_{S\bar{W}_{out}/S\bar{W}_{in}}$ | $\bar{\alpha}_{daily}$ | $LW_{in}$ | $LW_{out}$ | $T$ | $U$ | $RH$ |
|---|---|---|---|---|---|---|---|---|
| | $\left[\frac{W}{m^2}\right]$ | | | $\left[\frac{W}{m^2}\right]$ | $\left[\frac{W}{m^2}\right]$ | $[C]$ | $\left[\frac{m}{s}\right]$ | $[\%]$ |
| Bello | 297 | 0.26 | 0.28 | 231 (236) | 300 (306) | 2.3 | 2.9 | 37 |
| Pirámide | 282 | 0.07 | 0.07 | 267 | 362 | 7.0 | 4.0 | 40 |
| San Francisco | 211 | 0.36 | 0.37 | 261 (274) | 303 (318) | 7.1 | 2.0 | 43 |
| Mocho | 273 | 0.57 | 0.58 | | | 5.9 | 6.3 | 66 |
| Exploradores | 183 | 0.23 | 0.24 | 308 (349) | 310 (352) | 7.4 | 3.1 | 87 |
| Tyndall 2015 | 188 | 0.51 | 0.52 | 300 (314) | 314 (328) | 4.8 | 5.6 | 74 |
| Tyndall 2016 | 192 | 0.43 | 0.45 | 301 (315) | 314 (330) | 5.3 | 5.7 | 72 |

glaciological regions but also inside each region. In the Central Andes the AWSs installed on Bello and Pirámide Glacier receive considerably more incoming solar radiation $SW_{in}$ than the AWS on San Francisco Glacier which is receiving shade from Mirador del Morado peak in the morning hours (see Figures 1c and A1 in the supplementary material) The mean albedo of the surface was calculated by two methods: firstly by calculating for every day the quotient of the daily sum of outgoing divided by the sum daily incoming solar radiation and taking the average of these values ($\bar{\alpha}_{daily}$) and secondly by simply dividing the mean outgoing solar radiation by the mean incoming solar radiation over the study period ($\alpha_{S\bar{W}_{out}/S\bar{W}_{in}}$). Both methods give similar results. The heavily debris-covered Pirámide Glacier is showing very low albedo. Bello Glacier is showing an albedo expected for debris-rich ice and San Francisco Glacier is showing an albedo which can be associated to clean ice (Cuffey and Paterson, 2010). The incoming longwave radiation $LW_{in}$ is lower on Bello Glacier which can be explained by the lower atmospheric temperature due to its higher elevation (see Table 1.). The outgoing longwave radiation $LW_{out}$ is highest for Pirámide glacier, whose surface is heating considerably in the afternoons (see Figure 4 f and below). No bias correction could be applied to this data since we do not know the real surface temperature of the debris which covers the glacier surface. Bello and San Francisco show bias-corrected mean values of $LW_{out}$ of about 300 W/m$^2$. In the Central Andes wind speed is highest for Pirámide Glacier and the relative humidity is very similar for the three glaciers (around 40%).

At Mocho Glacier in the Lakes District, $SW_{in}$ is slightly lower than for Bello and Pirámide Glacier and the albedo is much higher, which is explained by the fact that on Mocho Glacier the AWS was installed near the ELA and snow or firn

were covering the glacier surface during the observation period (see Figure 9 and section 5.2). Both wind speed and relative humidity were clearly higher on Mocho Glacier in comparison with the glaciers of the Central Andes.

At the glaciers of the Patagonian Andes $SW_{in}$ is clearly lower than for the glaciers of the other regions. This can be explained by its latitudinal dependency, due to the higher absorption of the solar radiation in the more humid and cloudy atmosphere in the Wet Andes and due to the fact that the glaciers in Patagonia are located at lower elevations. The albedo is higher for the clean Tyndall Glacier as compared to the partly debris covered Exploradores Glacier (Figure 2). $LW_{in}$ is highest for Exploradores Glacier where also the highest relative humidity $RH$ is observed. At Tyndall Glacier (the bias corrected) $LW_{out}$ is very near to the expected value for a melting ice surface (315.6 W/m$^2$) in both years. For Exploradores Glacier it is slightly lower. Mean air temperature on Exploradores Glacier was similar to the one observed at Pirámide and San Francisco Glacier in the Central Andes and a bit lower at Tyndall Glacier. Measured wind speed was higher on Tyndall Glacier.

In order to study the daily cycle of the climatic variables on the glacier we calculated the average value which was measured at every hour of the day during the measurement period which are presented in Figure 4. As expected, the air temperature

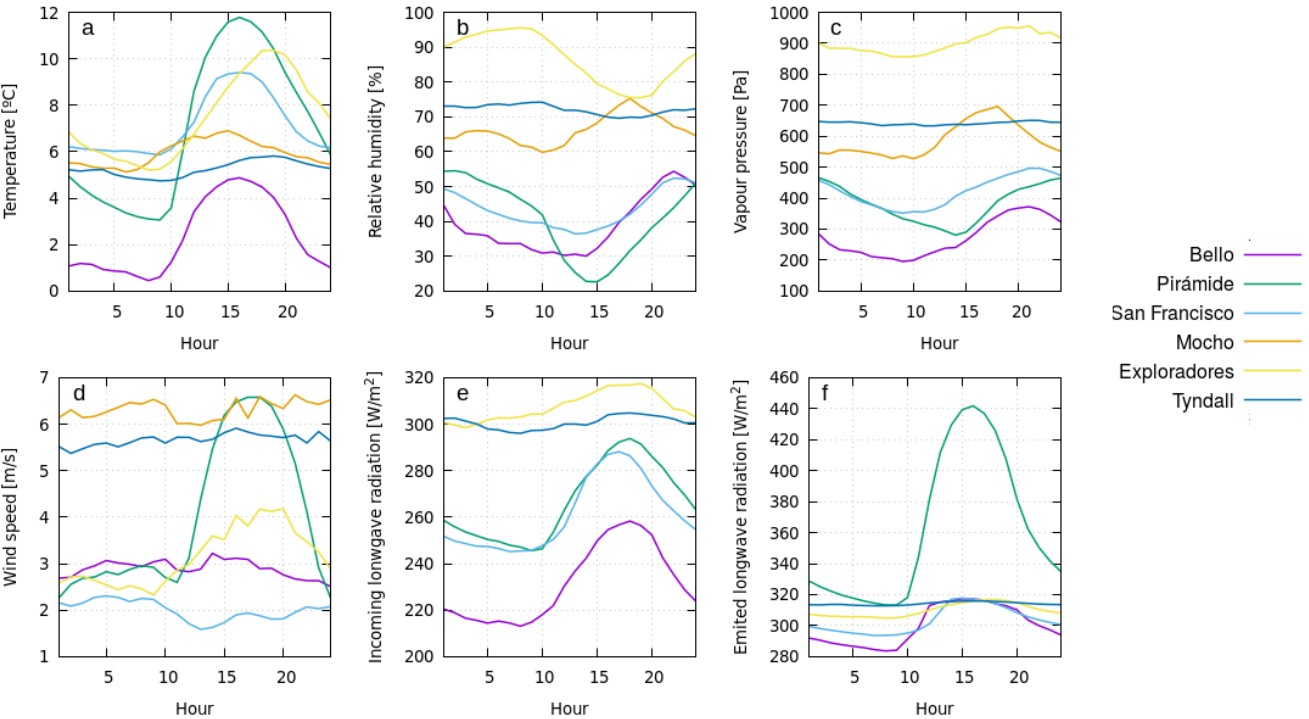

**Figure 4.** Averages of meteorological variables at the different hours of the day during the study periods on the six studied glaciers: a) temperature $T$, b) relative humidity $RH$, c) water vapor pressure $P_{vap}$, d) wind velocity $U$, e) incoming longwave radiation $LW_{in}$, f) outgoing longwave radiation $LW_{out}$.

shows a daily cycle for most of the glaciers, with maximum temperatures in the afternoon and minima in the early morning hours. However this daily cycle is much less pronounced for Tyndall and Mocho Glaciers. The relative humidity decreases

during the daytime for most of the glaciers. The water vapor pressure shows a maximum during the late afternoon, when the air temperature is still elevated and the humidity is increasing. The wind speed shows a very pronounced increase during the daytime at Pirámide Glacier, when its debris covered surface is heating (Figure 4f). At Exploradores Glaciers an increase in wind speed during the afternoon is observed as well. The incoming longwave radiation shows a maximum during the daytime, when the atmospheric temperature is highest. The outgoing longwave radiation, which is emitted by the glacier surface has a very distinct maximum during the afternoon for Pirámide Glacier (increase of more than 100 W/m$^2$), which means that the rock-covered glacier surface is warming during the daytime. Bello, San Francisco and Exploradores Glacier also experience a maximum emission of longwave radiation in the afternoon, whereas Tyndall Glacier shows a constant rate of emission of longwave radiation, which indicates a constant surface temperature during summer.

## 4.2   Average energy balance and melt

Since both EB-Model and COSIMA are models designed to compute the surface energy balance over snow and ice surfaces, we exclude the heavily debris covered Pirámide Glacier from the analysis of the surface energy fluxes. In Figure 5 and Table 5 we present the mean energy fluxes and inferred melt rates for the other five glaciers using the three different methods.

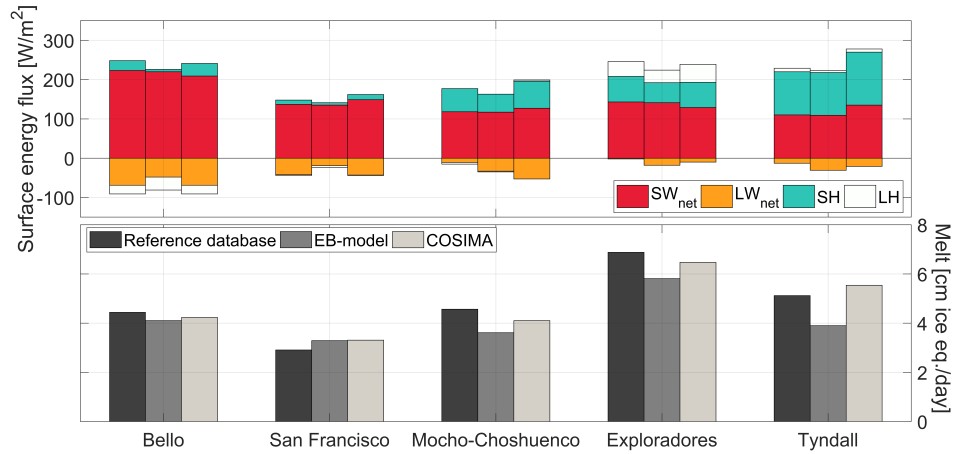

**Figure 5.** Mean modeled and measured energy fluxes and melt during summer for five glaciers: left bar Reference Database, middle bar EB-Model, right bar COSIMA.

In Figure 5 the three columns per glacier correspond to Reference Database, EB-Model and COSIMA from left to right. For Tyndall Glacier only the results for the summer season 2016 are shown. The net energy flux towards the glacier was converted into daily melt rates in ice equivalent using an ice density of 917 kg/m$^3$. The mean pattern of the energy fluxes changes from the Central Andes to Patagonia. The net shortwave radiation decreases from North to South. The net longwave radiation is negative in the Central Andes and near to zero in Patagonia. The sensible heat flux is a more important source of energy in Patagonia. The latent energy flux changes sign form sink of energy in the Central Andes to source of energy in Patagonia.

**Table 5.** Mean values of the computed surface energy fluxes and melt rates during the study periods on the five studied glaciers using the three different methods. For the Reference Database turbulent fluxes and melt rates calculated without using stability corrections are indicated in parenthesis.

| **Glacier** | Method | $SW_{\mathrm{net}}$ | $LW_{\mathrm{net}}$ | $SH$ | $LH$ | Melt |
|---|---|---|---|---|---|---|
| | | $\left[\frac{W}{m^2}\right]$ | $\left[\frac{W}{m^2}\right]$ | $\left[\frac{W}{m^2}\right]$ | $\left[\frac{W}{m^2}\right]$ | [cm ice eq./day] |
| Bello | Reference Database | 223 | -69 | 25(32) | -22(-29) | 4.4 (4.4) |
| | EB-Model | 220 | -48 | 6 | -33 | 4.1 |
| | COSIMA | 208 | -69 | 32 | -22 | 4.1 |
| San Francisco | Reference Database | 137 | -42 | 11(41) | -2(-9) | 2.9 (3.6) |
| | EB-Model | 135 | -19 | 6 | -5 | 3.3 |
| | COSIMA | 149 | -43 | 13 | -1 | 3.3 |
| Mocho | Reference Database | 118 | -11 | 59(74) | -4(-5) | 4.6 (5.0) |
| | EB-Model | 117 | -33 | 46 | -2 | 3.6 |
| | COSIMA | 127 | -53 | 69 | 3 | 4.0 |
| Exploradores | Reference Database | 143 | -2 | 65(94) | 38(55) | 6.9 (8.2) |
| | EB-Model | 141 | -18 | 51 | 32 | 5.8 |
| | COSIMA | 129 | -10 | 64 | 46 | 6.4 |
| Tyndall 2015 | Reference Database | 94 | -14 | 65(80) | 9(10) | 4.5 (4.8) |
| | EB-Model | 92 | -30 | 52 | 5 | 3.3 |
| | COSIMA | 132 | -22 | 70 | 9 | 5.3 |
| Tyndall 2016 | Reference Database | 110 | -13 | 76(87) | 9(10) | 5.1(5.5) |
| | EB-Model | 109 | -29 | 55 | 5 | 3.9 |
| | COSIMA | 135 | -21 | 75 | 8 | 5.5 |

This means that in Patagonia water vapor condensates at the surface of the Glaciers, which generates heat for additional melt. There are differences in the prediction of the energy fluxes on the Glacier surfaces between the different methods which will be discussed in detail in the next section. The predicted melt rates for the specific study points (locations of the AWS) in the ablation area of the glaciers are higher for the Patagonian Glaciers as compared to the glaciers of the Central Andes.

5  **4.3  Daily energy balance and melt**

In the Figures 6,7,8, we present the computed daily energy fluxes and melt rates for Bello Glacier, Mocho Glacier and Tyndall Glacier (2016) respectively. On Bello Glacier the melt rates are clearly modulated by the net shortwave radiation: on days with reduced net shortwave radiation (due to the presence of clouds) , melt rates show minima (Figure 6). On Mocho Glacier this

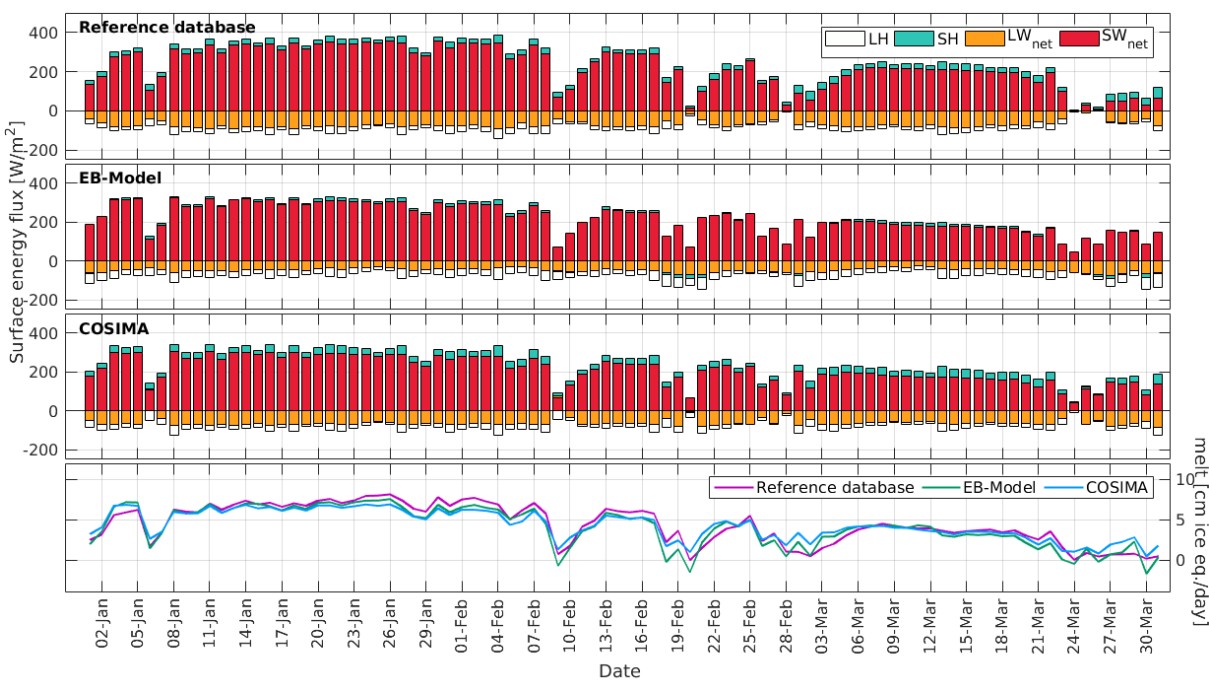

**Figure 6.** Daily modeled and measured energy fluxes and inferred daily melt rates during summer 2015 for Bello Glacier using the three methods.

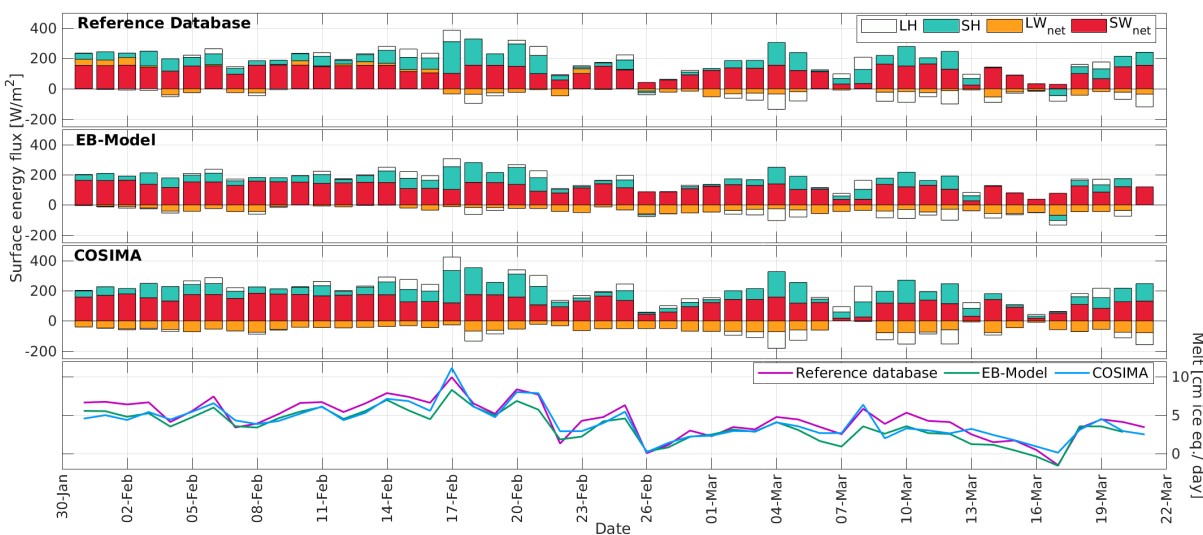

**Figure 7.** Daily modeled and measured energy fluxes and melt during summer 2006 for Mocho Glacier using the three methods.

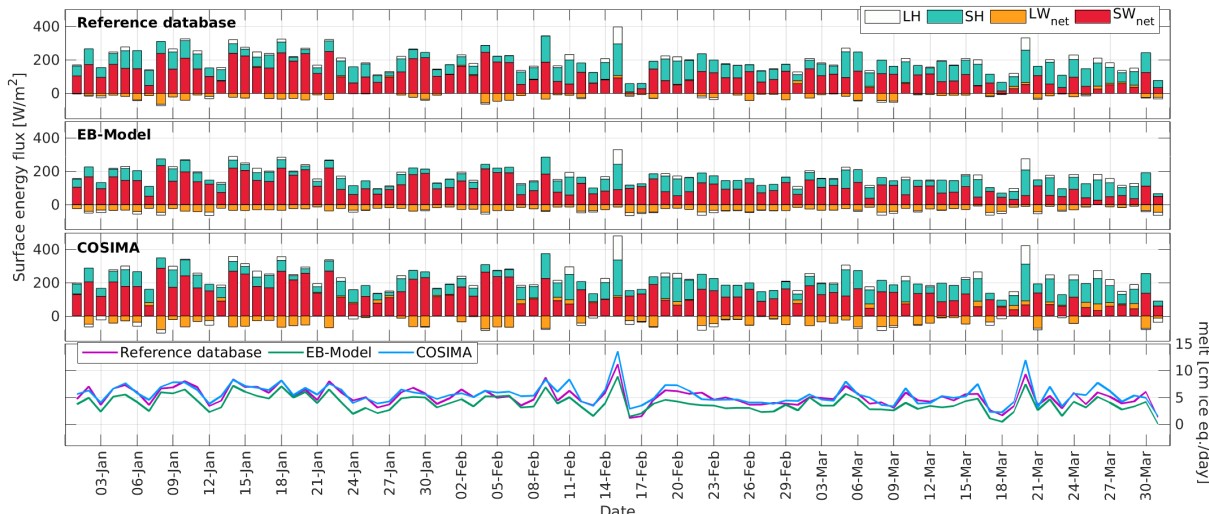

**Figure 8.** Daily modeled and measured energy fluxes and melt during summer 2016 for Tyndall Glacier using the three methods.

picture changes: high melt rates are rather associated with low net solar radiation, high contributions of the sensible heat flux and positive values of the latent heat flux (see for example 17th of February or 8th of March in Figure 7).

Similar results can be observed for Tyndall Glacier: peaks in melt rates are associated to low contribution of the net shortwave radiation and high contributions of the turbulent heat fluxes (15th of February or 20th of March in Figure 8).

Statistics of the comparison of the modelled and measured energy fluxes and the modelled melt rates by the three methods are presented in Table A1. A comparison of the hourly modelled and measured energy fluxes during the first three days of february on Bello Glacier and Tyndall Glacier are presented in Figure A2.

## 5   Discussion

### 5.1   Microclimatic conditions on the glacier surface

The systematic variation of several meteorological variables between the different parts of the Chilean Andes crucially determine the importance of the different energy exchange process at the glacier surfaces (Table 4). $SW_{\text{in}}$ differs around 100 W/m$^2$ form the Central Andes (considering Bello and Pirámide Glaciers) to the Patagonian Andes. The difference in $SW_{\text{in}}$ between the two glaciers in the Patagonian Andes is only 9 W/m$^2$, although they have opposite exposition. The difference between two summers on Tyndall Glacier is only 4 W/m$^2$. Another clear trend form north to south is found for the relative humidity. The

values measured in the Patagonian Andes double the values obtained for the central Andes. This influences the latent fluxes: in the Central Andes moisture is transported away from the glacier surfaces whilst in Patagonia moisture is transported towards the glacier surfaces. Although the mean air temperature measured over Pirámide, San Francisco and Exploradores Glaciers were very similar, the incoming longwave radiation was much higher ($>75\,W/m^2$) at Exploradores Glacier. This can be ex-

plained by a higher emissivity of the atmosphere due to a higher relative humidity of the air and due to more presence of clouds in the humid conditions of Patagonia.

Considering the variability of the data from the two summers measured on Tyndall Glacier, we can state that the glacier climate was similar in both summers. Especially mean $LW_{in}$, $LW_{out}$, $RH$ and $U$ were nearly identical. $SW_{in}$ and $T$ were slightly lower in 2015 as compared to 2016 and the surface albedo was higher in 2015. The mean values of $T$, $RH$ and $U$ measured on Tyndall Glacier during the summers 2015 and 2016 were also very similar to the mean values measured by Takeuchi et al. during December 1993 (Takeuchi et al., 1999).

## 5.2 Parametrizations of the surface energy fluxes

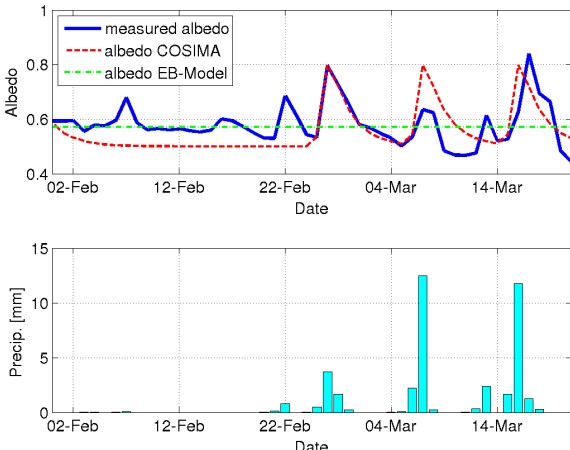

**Figure 9.** Measured and modeled daily albedo on Mocho Glacier and precipitation registered in Puerto Fuy during summer 2006. The albedo value used in EB-Model corresponds to the mean value of the measured albedo. In COSIMA the following parameters have been chosen: $\alpha_{frsnow} = 0.8$, $\alpha_{firn} = 0.5$, time constant $t^* = 2$ days, snow depth constant $d^* = 8$ cm (see equations (10) and (11))

The net shortwave radiation is an important source of energy for all the glaciers. According to equation (5) it is determined by the incoming shortwave radiation and the albedo of the surface. Albedo of snow and ice surfaces are very variable and depend on grain size and form, liquid water content, impurities and other factors (Wiscombe and Warren, 1980; Warren and Wiscombe, 1980; Cuffey and Paterson, 2010). Generally fresh snow has the highest albedo which is decreasing in time when snow grains are growing and the snow is eventually getting dirty. COSIMA tries to reproduce this albedo aging effect introducing a snow albedo which exponentially decreases in time (equation 11). In Figure 9 we show the comparison between the measured daily albedo on Mocho Glacier during February and March 2006 and the predictions of the COSIMA model. In comparison to other studies that used similar albedo parametrizations (Mölg et al., 2012; Huintjes et al., 2015b), we only sligthly reduced the albedo of fresh snow and firn (by 0.05), but significantly reduced the time constant $t^*$ by a factor 3. Most of the measured increases in the albedo can be associated to precipitation events registered at the nearby automatic weather station in Puerto Fuy (Figure

9). COSIMA is able to capture these increases, however, the measured increases in the surface albedo are much more variable than the ones obtained from the model. A drawback of this comparison is certainly that we do not know the exact amount of snow falling on the glacier, but deduce it from the liquid precipitation measured at a automatic weather station in the valley. The faster reduction of the albedo after snowfall and the associated lower value of the time constant $t^*$, indicates a faster snow metamorphism on Mocho Glacier during the observation period, probably due to higher ambient temperatures in comparison to the high-altitude sites studied in Mölg et al. (2012) and Huintjes et al. (2015b). Considering the statistics of the modeled daily net shortwave radiation (Table A1), it is important to note that the slightly lower root-mean-square-deviation for EB-Model was obtained by the mean measured albedo, whilst in COSIMA a standard albedo of ice of 0.3 was applied.

The longwave radiative fluxes make important contributions to the energy exchange at the glacier surface. Since snow and ice emit approximately like blackbodies ($\varepsilon = 1$) in this part of the electromagnetic spectrum and the atmosphere mostly shows emissivity smaller than one, the longwave radiation balance is often negative (even at positive ambient temperatures). However in the humid Patagonia the measured net longwave radiation is often small (Figure 5 upper panel left bar glaciers Mocho, Exploradores and Tyndall). In Figure 10 we show the "measured" daily emissivity calculated by inverting the Stefan-Boltzmann law

$$\varepsilon_{\text{measured}} = \frac{LW_{\text{in}}}{\sigma T^4}, \tag{14}$$

where $\sigma$ denotes the Stefan–Boltzmann constant, as a function of the daily cloudiness for the glaciers Bello, San Francisco Exploradores and Tyndall. For a direct comparison, we show the parametrizations of the models in the plots of every glacier. Generally we can note that the data of the "measured" emissivity shows considerable scatter around their trend line. This may indicate that the variability of the measured emissivity can not only be explained by the variability of the cloudiness. Probably the relative humidity and other factors will influence the emissivity of the atmosphere. However the big scatter of the measured data might also be associated with the uncertainties of the determination of the cloud cover and the emissivity of the atmosphere. The cloud cover data were obtained from specific parametrizations of the transmissivity of the atmosphere as a function of the cloudiness. However these parametrizations are not unique (see for example Oerlemans (2001)). Also the "measured" emissivity is associated to some uncertainty: it is not clear if the temperature measured at two meters over the glacier surface is representative for the temperature of the atmosphere which is emitting longwave radiation towards the glacier surface. We can recognize that the model parametrizations underestimate the emissivity of the atmosphere at Exploradores Glacier for low cloudiness conditions. This underestimation of the emissivity of the atmosphere leads to an underestimation of the net longwave radiative balance on clear days. At San Francisco Glacier the model parametrizations overestimate the emissivity of the atmosphere for cloudy conditions. Considering the statistics of the modeled daily net longwave radiation (Table A1), we can recognize that the correlation is much lower than in the case of the modeled net shortwave radiation. Especially the negative correlations in the case of EB-Model for the glaciers of the Central Andes are indicating that here this model is not able to reproduce the measured daily variations of this flux. Reasons for that could be a not successful determination of the cloudiness conditions by this method and the fact that during nighttimes a constant cloudiness was used.

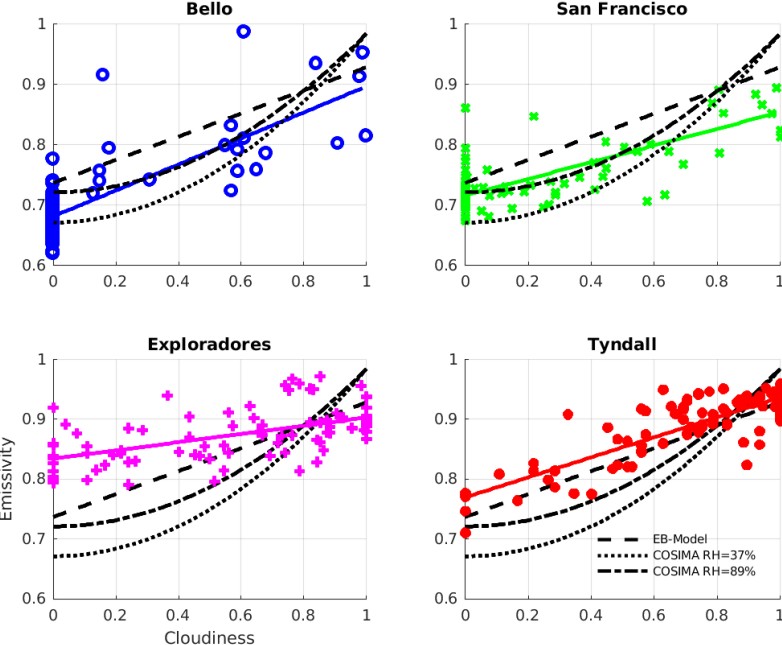

**Figure 10.** "Measured" emissivity of the atmosphere as a function of the cloudiness. The data points correspond to the daily emissivity obtained from equation (14) by using daily means of $LW_{in}$ and $T$ plotted against the inferred daily cloudiness values. The continuous colored straight lines correspond to linear fits to the data of every glacier and the black discontinuous lines correspond to the model parametrizations ( same as Figure 3 b).

The variability of the modeled turbulent fluxes is very similar in all three methods (see Table A1). This is expected since the formula to compute these fluxes have very similar aspect: in all approaches the sensible heat flux is mainly driven by the temperature difference of the glacier surface and the atmosphere at two meters elevations and wind speed and the latent heat flux is driven by the difference of the water vapor pressure at the glacier surface and the atmosphere at two meters elevations and wind speed. The mean values of the turbulent fluxes computed in EB-Model are lower than the one obtained by the other two methods (Table 5 and Figure 5). This is because the EB-Model assumes a glacier surface at zero degrees Celsius which reduces both the temperature difference between glacier surface and the overlying air layer and the difference of water vapor content between both. This causes the melt rates modeled by EB-Model to be generally lower (Figure 5). The modeled turbulent fluxes in the Reference Database and by COSIMA are very similar, except for Mocho Glacier where a glacier surface at zero degrees had to be assumed in the Reference Database, due to the lack of data of the outgoing longwave radiation.

In Table 5, for the Reference Database, we also present in parenthesis values obtained for the turbulent fluxes without applying a stability correction and the resulting inferred melt rates. Important differences can be noted especially for the glaciers where the mean wind speed is moderate (Exploradores, San Francisco and Bello). The strongest influence of the

stability correction on the melt rate is for Exploradores Glacier, because both turbulent fluxes have the same sign, The stability correction for the sensible heat flux and the latent heat flux perfectly cancel out at Bello Glacier. The influence of the surface roughness on the turbulent fluxes has a similar character: doubling $z_0$ for Bello Glacier will change the net energy flux by only 1 W/m$^2$ and using a roughness length of 10 mm (twenty times higher) by 5 W/m$^2$. At the Patagonian glaciers, however, a higher surface roughness would have a much higher impact since both turbulent fluxes have the same sign.

### 5.3 Melt rates

Mean melt rates ranging from 2.9 to 4.4 cm/day of ice equivalent (cm i.e./day) for the Dry Andes and from 3.3 to 6.9 cm i.e./day for the Wet Andes were predicted by the different approaches of quantifying energy fluxes on the surface of five glaciers (Table 5). These values lie within the range of observed melt rates during summer on these glaciers or glaciers with similar climatic conditions.

In the Wet Andes Schaefer et al. (2017) measured ablation rates of 2.6 cm i.e./day and 3.2 cm i.e./day in the summers 2010 and 2011 and measured and inferred rates of 3.6 cm i.e./day and 3.7 cm i.e./day in the summers 2012 and 2013 on Mocho Glacier at the same location where the AWS was installed in 2006. This indicates that the Reference Database and COSIMA may overestimate the melt at this location. Here, we have to take into account that for Mocho Glacier net longwave radiation was inferred by subtracting the net shortwave fluxes from the net allwave radiation, measured with the NR-lite, which is a lower precision instrument as compared to the newer sensors installed in the CNR4. At Tyndall Glacier in the period November 2012 to May 2013 an average ablation rate of 3.8 cm i.e./day was observed at two stakes near to the location of the AWS. Considering that this period also includes spring and autumn months, were melt rates should be lower, this in in good agreement to the values of 3.3 to 5.1 cm i.e./day predicted by the different approaches presented in this work. The modeled melt rates for the Wet Andes are also in agreement with the melt rates of 4-5.5 cm i.e./day observed during summer on Perito Moreon Glacier on the Southern Patagonian Icefield (Stuefer et al., 2007) and the 4-8 cm i.e./day observed at Nef Glacier (Schaefer et al., 2013), both at elevations of about 500 m.a.s.l.. From January 2015 to March 2015 an average ablation rate of 9.1 cm i.e./day was measured at a stake network installed on Grey Glacier at an elevation range ranging from 260 m.a.s.l. to 380 m.a.s.l. . This indicates that the high melt rates modeled for Exploradores Glacier seem to have a realistic magnitude for a low elevation site on a glacier in the Patagonian Andes, although the different climate conditions at Grey Glacier make a direct comparison difficult.

In the Dry Andes ablation was measured at a stake network on Bello Glacier during summers 2013/2014 and 2014/2015 (CEAZA, 2015). A high variability of ablation rates in space and time were obtained. Several of the ablation rates inferred for stakes nearby the AWS were of similar size than the predicted ones by the methods presented in this paper. Analyzing the signal of an ultrasonic sensor installed on an ablation gate next to the AWS of San Francisco Glacier we found a surface lowering of 4.9 cm/day during December 2015. Assuming that snow melt in this period of the year with a density of 500 kg/m$^3$ this yields a rate of 2.7 cm i.e./day, which is in good agreement with the melt rates inferred in this contribution.

When comparing modeled melt rates between the three methods presented in this study, we can state that there exist considerable differences between the three approaches with a root-mean-square-deviation of the daily modeled melt rates of around one cm ice eq./day (see Table A1).

## 5.4 Implications for glaciological zones in Chile

Our results suggest a transition of energy sources for surface melt from the Central Andes to Patagonia. The energy sources obtained for Mocho Glacier at $40°$S are more similar to the ones observed at the Patagonian glaciers than to the ones observed for the glaciers of the Central Andes, where $SW_{in}$ is the dominating source of energy for melt (Figures 5 and 6). The greater importance of the turbulent flux of sensible heat as energy source available to produce surface melt at Mocho Glacier probably contributes to the observed strong dependency of its annual mass balance on the annual mean temperatures measured on a
nunatak of the glacier(Scheiter, 2016; Schaefer et al., 2017). This picture changes strongly in the Central Andes, where a very low dependency of the annual mass balance of Echaurren Norte Glacier on the annual mean temperature observed at Embalse el Yeso was reported (Masiokas et al., 2016; Carrasco, 2018; Farías-Barahona et al., 2019). When comparing the importance of the energy sources between the two summers modeled for Tyndall Glacier (Table 5) , we can state that in 2015 $SW_{net}$ was slightly lower than in 2016 due to the lower $SW_{in}$ and the higher surface albedo observed in that year (Table 4). However the
overall pattern of energy sources (and sinks) is very similar for both years.

In Figure 11a we plot the predicted melt rates by the Reference Database against the mean air temperature for the six

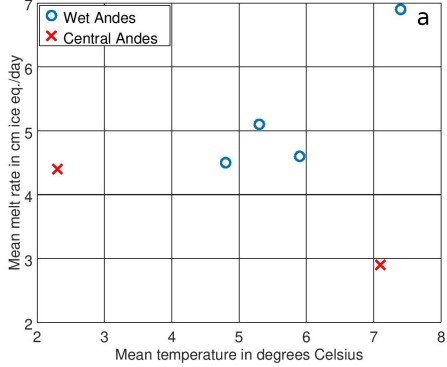
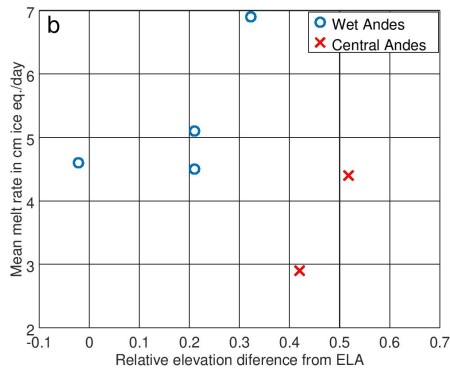

**Figure 11.** Melt rate in the different zones as a fucntion of temperature (a) and as a function of the the relative elevation difference to the ELA (b).

modeling periods (two periods for Tyndall Glacier) and in Figure 11b against the relative elevation difference of the AWS with respect to the equilibrium line altitude (ELA), which we define as: (ELA-ElevationAWS)/(elevation span glacier). For the glaciers of the Wet Andes we can see a clear increase of the melt rates as a function of the mean temperature. For the
modeled glaciers of the Central Andes, however this trend does not exist. In both regions the melt rates increase with the

relative elevation difference to the ELA, however at similar relative elevation difference the glaciers of the Wet Andes show clearly higher melt rates.

All these results confirm the general division of the Chilean Andes into Dry Andes and Wet Andes, with the zonification limit being located approximately at 35°S (Lliboutry, 1998).

## 5.5 Implications for physical melt modeling and transferability of parametrizations

The capacity of the models to reproduce the measured radiative fluxes can still be improved. The albedo aging effect implemented in COSIMA is a big improvement compared to constant albedo parametrizations for snow, firn and ice surfaces. However the parameters of this aging formula seem to vary strongly from one site to another and a good calibration of this formula seems to be necessary. Regarding the predictions of the net longwave radiative fluxes on the glacier surface, the parametrization of the emissivity of the atmosphere is crucial. The tested models cannot reproduce the variability of the emissivity as a function of the cloudiness for all the glaciers. Especially at Exploradores Glacier the clear-sky emissivity is underestimated by the models. This is because the used parametrizations are fits to data that were obtained in different climatic conditions. Therefore, these parametrizations are not physical and can not be simply transferred to other sites where the conditions are different.

Different parametrizations for the turbulent fluxes of sensible and latent heat were compared in this study. The transfer coefficient depends directly on the roughness length $z_0$ (and $z_T$, $z_H$ in the case of EB-Model), which is/are therefore crucially determining the magnitude of the turbulent fluxes. However these roughness lengths are neither constant in time nor in space, which makes them very difficult to determine. A common practice is to chose the roughness length in a way that the modeled melt rates agree with the measured ones. However, this exercise does not make these formula very adequate predictors of melt rates in other situations and on other glacier surfaces. More direct measurements of turbulent fluxes over glacier surfaces (for example using the eddy-covariance technique (Cullen et al., 2007; Litt et al., 2015)) are necessary to find physical parametrizations of these fluxes.

Generally, there is still a strong need of measurements of the energy exchange processes over glacier surfaces and we think that coordinated efforts of governmental agencies, such as the Glaciology and Snows Unit of the Chilean Water Directory in our case, can make important contributions. If we want to work towards physical melt models, then we have to test the capacity of the models to reproduce the different physical processes that take place at the glacier surface.

## 6 Conclusions

Performing an extended study of surface energy fluxes during summer on five Chilean glaciers on a north-south transect and under strongly varying climate settings we reached to the following conclusions:

- The contribution of the different surface energy fluxes over glacier surfaces change from the Central Andes towards the Patagonian Andes: the net shortwave radiation as main source of energy in the Central Andes looses importance further South, where the turbulent fluxes of sensible and latent heat are providing more energy for melt. The net longwave

radiation changes from a strong sink of energy for the glaciers of the Central Andes to a net zero contribution in the Patagonian Andes.

– For the glaciers of the Wet Andes a clear dependency of the modeled melt rates on the mean air temperature was observed. This dependency did not exist for the studied glaciers in the Central Andes.

– The inferred melt rates increased with the relative elevation diference from the ELA in both studied glaciological zones. The modeled melt rates were higher for the Patagonian Andes than for the Central Andes.

– Mocho Glacier in the Chilean Lake District is showing similar patterns of surface energy fluxes to the glaciers in the Patagonian Andes.

– The models underestimated the emissivity of the clearsky atmosphere at Exploradores Glacier, an extremely humid place in the Wet Andes.

– From our study it is difficult to infer which parametrization of the turbulent fluxes is the most appropriate one. More detailed studies on this topic are necessary, which include direct measurements of these fluxes.

– To develop or improve physical melt models, we have to validate every single model parametrization against data and cannot judge the model's performance only by the final output. In this highly parameterized models the effect of physically wrong parametrizations might cancel out and the final result might be satisfying, without reproducing well the individual physical processes.

– Openly shared codes are the best way to improve physical models, since everyone can test the individual parametrizations against his data and adjust or improve them accordingly. This is the preferred way to improve physical parametrizations as opposed to large chains of models which try to model physical processes without validating intermediate model results.

*Author contributions.* M.S. designed the research, ran the COSIMA model, wrote the manuscript and prepared several figures, D.Fonseca prepared the Reference Database, ran EB-Model and prepared several figures, D.Farías provided detailed information about the DGA-AWS and prepared Figure 1, G.C. measured the surface energy fluxes on Mocho Glacier. All authors discussed the results and commented on the manuscript.

*Code availability.* Eb-Model is freely available at:
https://onlinelibrary.wiley.com/doi/abs/10.1002/1096-9837%28200006%2925%3A6%3C649%3A%3AAID-ESP97%3E3.0.CO%3B2-U and COSIMA is available at https://bitbucket.org/glaciermodel/cosima/src/master/ and the newer version implemented in python at:
https://github.com/cryotools/cosipy (recomended).

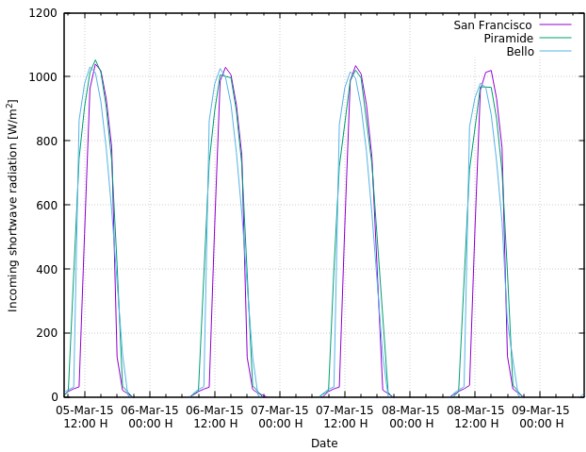

**Figure A1.** Incoming shortwave radiation at the AWSs installed on Bello, Pirámide and San Francisco Glacier during March 2016.

**Table A1.** Statistical comparison of the modeled daily energy fluxes and melt by EB-Model and COSIMA with the reference database. Pearson correlation coefficient ($r$) and root-mean-square-deviation ($RMSD$) are indicated.

| Quantity | Statistic | Bello | San Francisco | Mocho | Exploradores | Tyndall 2015 | Tyndall 2016 | Mean |
|---|---|---|---|---|---|---|---|---|
| $SWr_{net}$ | $r_{EB}$ | 0.92 | 0.99 | 0.91 | 0.98 | 0.89 | 0.97 | 0.94 |
| | $r_{COS}$ | 0.92 | 0.99 | 0.92 | 0.98 | 0.89 | 0.97 | 0.94 |
| | $RMSD_{EB}$ | 46 | 9 | 18 | 13 | 24 | 16 | 21 |
| | $RMSD_{COS}$ | 50 | 15 | 20 | 21 | 47 | 29 | 30 |
| $LW_{net}$ | $r_{EB}$ | -0.51 | -0.09 | 0.49 | 0.52 | 0.35 | 0.60 | 0.23 |
| | $r_{COS}$ | 0.65 | 0.76 | 0.36 | 0.69 | 0.76 | 0.81 | 0.67 |
| | $RMSD_{EB}$ | 34 | 29 | 32 | 22 | 25 | 22 | 27 |
| | $RMSD_{COS}$ | 16 | 13 | 50 | 26 | 20 | 24 | 25 |
| SH | $r_{EB}$ | 0.14 | 0.80 | 0.99 | 0.99 | 0.96 | 0.96 | 0.81 |
| | $r_{COS}$ | 0.79 | 0.86 | 0.98 | 0.99 | 0.98 | 0.98 | 0.93 |
| | $RMSD_{EB}$ | 22 | 7 | 18 | 24 | 22 | 24 | 19 |
| | $RMSD_{COS}$ | 10 | 4 | 15 | 8 | 8 | 8 | 9 |
| LH | $R_{EB}$ | 0.72 | 0.29 | 1.00 | 0.99 | 0.96 | 0.98 | 0.82 |
| | $R_{COS}$ | 0.87 | 0.23 | 0.98 | 0.99 | 0.97 | 0.98 | 0.84 |
| | $RMSD_{EB}$ | 14 | 6 | 8 | 13 | 7 | 6 | 9 |
| | $RMSD_{COS}$ | 5 | 6 | 11 | 13 | 9 | 10 | 9 |
| Melt | $r_{EB}$ | 0.95 | 0.96 | 0.96 | 0.97 | 0.92 | 0.94 | 0.95 |
| | $r_{COS}$ | 0.95 | 0.94 | 0.92 | 0.95 | 0.94 | 0.91 | 0.93 |
| | $RMSD_{EB}$ | 0.8 | 0.6 | 1.2 | 1.4 | 1.5 | 1.4 | 1.1 |
| | $RMSD_{COS}$ | 0.93 | 0.6 | 1.0 | 1.1 | 1.1 | 0.9 | 0.9 |

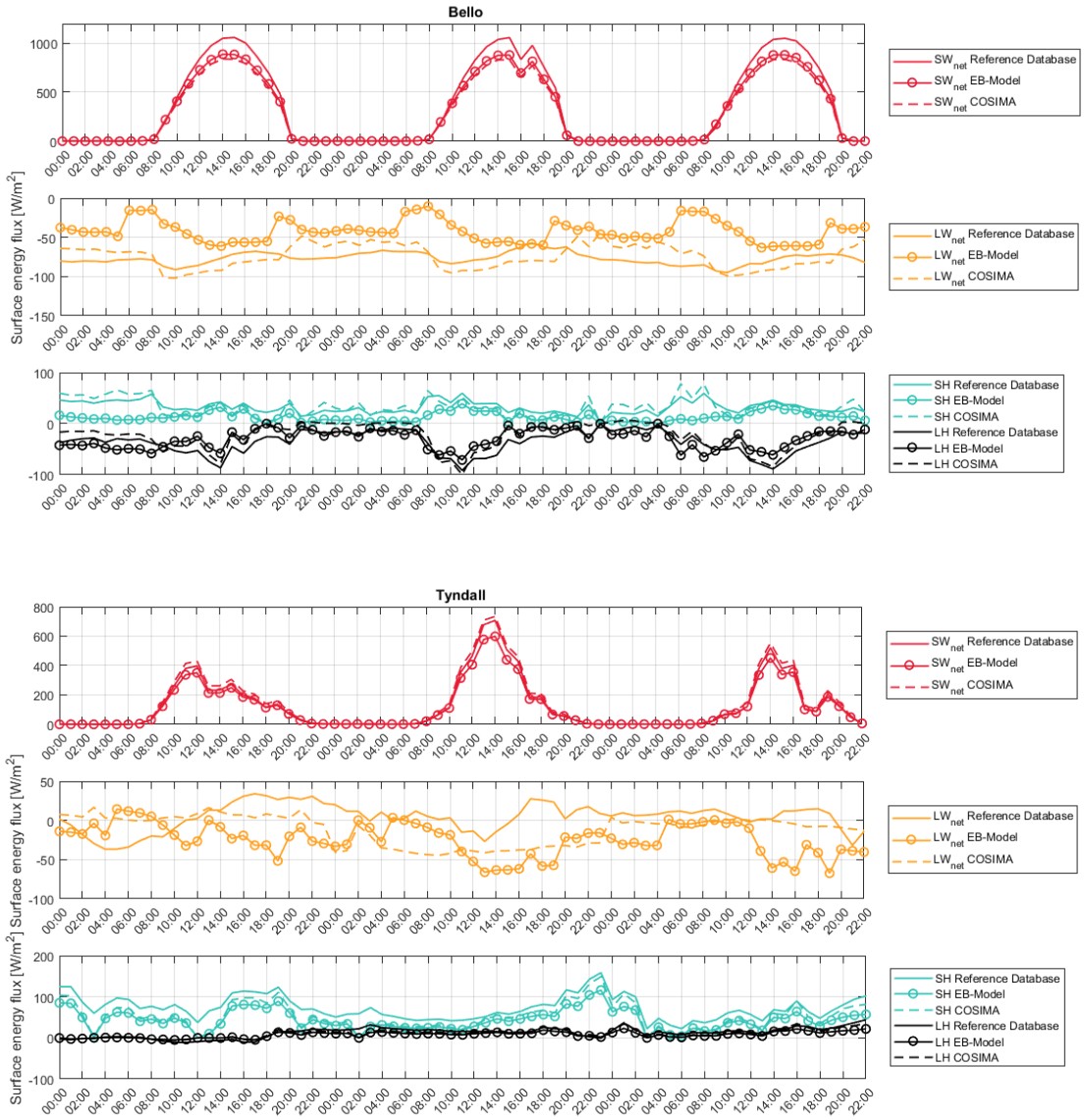

**Figure A2.** Comparison of modelled and measured hourly energy fluxes on glaciers Bello (top) and Tyndall (2016, bottom) for the first three days of February.

*Data availability.* Input and output data presented in this manuscript can be obtained by solicitude to the corresponding author.

*Competing interests.* All authors declare no competing interests.

*Acknowledgements.* We thank the Chilean Water Directory (DGA) for providing meteorological information and photographs from the automatic weather stations according to "Ley de transparencia solicitud Numbers 76796,76800,76802". D F-B acknowledges the support
5   from ANID through the Chilean scholarship program. This research was supported by the FONDECYT Regular Grant No. 1180785.

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
