# Peer review of "Surface energy fluxes on Chilean glaciers: measurements and models"

_The Cryosphere, 2019_

## Referee Comment (RC1) · Anonymous Referee #1 · 10 Jul 2019

Comments for the Authors

The paper addresses relevant scientific questions within the scope of TC. It shows a good data set of meteorological measurements and estimates of the surface energy balance of Chilean glaciers of contrasting climates. Unfortunately, in my opinion the manuscript does not represent substantial progress beyond current scientific understanding. The overall presentation is well structured and clear. However, the text can be improved; it should be more concise and accurate. I found that some scientific methods were not suitable and that some assumptions were not clearly outlined. Finally, the discussion does not reach substantial conclusions. In my opinion, the main problem is that the objectives of the study are not clearly defined. The manuscript presents various results but the overall purpose of the study remains imprecise. The analysis of this

interesting dataset could be an interesting contribution to the literature if the objectives were articulated more clearly and reflected in an appropriate methodology. The paper would certainly need major improvements to be innovative and merit publication in TC.

These points are detailed in the comments below.

Major comments

- The objectives should be redefined.

The main contribution of the study is the presentation of a comprehensive set of meteorological measurements on glaciers of contrasting climates along the Andes. The data certainly allow for correct estimation of the energy fluxes at the surface of the glaciers. Radiation flux measurements seem appropriate and accurate. Turbulent flows are not measured directly, but meteorological measurements probably allow correct estimates (if the appropriate methods are applied, see comments below). Thus, I would suggest focusing on a comparison of the energy fluxes partitioning in the different climates from the dry Andes to the wet Andes (the measurements above debris-covered ice are not useful here). The effects of latitude and of altitude on the energy fluxes need to be discussed in more detail.

One limitation of the study is that general interpretations of different climates are deduced from point-scale measurements. The partitioning of the energy fluxes depends on the position of the weather station. For instance, the albedo varies greatly over short distances near the snow line, so that the interpretation of punctual energy flux measurements can lead to an erroneous generalization of the melting characteristics to the entire ablation area. This point should be discussed.

- Some applied methods are not suitable

The NR-Lite sensor is less accurate than the CNR4. Thus, calculations of longwave radiation fluxes can be problematic on the Mocho glacier.

Comparing different models does not bring much newness here. Some assumptions

are not valid (Ts=0°C, constant albedo…) or some formulations are not adequately described (see below). I would suggest discussing the energy fluxes derived from the most direct approach: the 'reference database' based on measurements. For example, there is no need to assume that the surface temperature is fixed to 0°C (P7, line 14) if outgoing longwave measurements are available. This assumption (which does not seem valid on the San Francisco and on Bello glaciers) has a significant impact on the turbulent sensible heat fluxes derived from the bulk method.

P8, Equation 4: use the standard relationship of saturation vapor pressure as a function of temperature, no need to test different parametrizations.

P8, line 13: why mentioning direct and diffuse components of solar irradiance if global radiation is directly measured?

Sections 3.2, 3.3 and 3.4: the comparison of the different turbulent 'transfer coefficient' (P7, line 16) should refer to stability concepts. A stability correction must be included over glacier surfaces, using the Monin-Obukhov length scale or the Richardson number. This important point should be clarified. The values of the roughness lengths for momentum, temperature and humidity should also be discussed in more detail with reference to the state of art.

P11, lines 9-12: no need to compare the two methods (especially if they give 'similar results').

The effects of cloud cover in the different climates should be investigated more rigorously. The applied method is inappropriate (the text does not say how the cloud cover is calculated) and appropriate references are missing. Figure 10 is unclear and its interpretation P17-18 remains vague. Robust methods have been proposed for estimating cloud cover from solar and longwave radiation fluxes measurements (e.g., Marty et al., TAC, 2002 in the Alps; Sicart et al., JOG, 2010 in the tropical Andes; McDonell et al., TAC, 2013 in the semiarid Andes of Chile; Munneke et al., IJC, 2011…). These parametrizations, once calibrated at each site, will make it possible to distinguish clear

skies from cloudy conditions. This point, with an adequate methodology, could be an interesting contribution of the study.

- Many results are presented without proper interpretation

Figures 5, 6, 7 and 8 show many results, but most of them remain poorly analysed.

The turbulent fluxes are not measured directly, so their estimates are not very accurate. The large differences in sensible heat flux in the different climates are certainly significant. However, I think no much can be said about the latent heat fluxes shown in figure 5; the fluxes remain close to zero and the uncertainties are certainly large.

- The text must be carefully proofread.

Be more specific and accurate, for instances: - P2, lines 23-25: give numbers to quantify the trends in mass balance - P5: the correct terms are 'irradiance' or 'radiation fluxes' (in W/m2) - P13, line 6 'ice equivalent'? Do you mean water equivalent? - P13 where are the figures 12 and 13? - P16, line 16: Equation 12? - Where is Table 4?

Minor comments

- P16, lines 2-4: the effects on longwave radiation of the "very humid and temperate air column between the sensor and the glacier surface" is probably small and can be estimated [e.g., Pluss and Ohmura, 1997].

- P19: why not using the meteorological and ablation measurements on the Bello glacier during the summers 2013/14 and 2014/2015 to validate (in a rigorous way) the calculations of the energy fluxes?

---

## Referee Comment (RC2) · Anonymous Referee #2 · 11 Nov 2019

This paper focuses on calculations of the surface energy balance (SEB) of glaciers along the Andes of Chile. The analysis is covering a large latitudinal range, between 18°S and 55°S, and tries to describe the main differences existing in the processes controlling melting under diverse climate settings. The paper compares three different modelling approaches applied on a dataset of 6 glaciers. The authors intend determining adequate parameterization of SEB models and conclude that the use of observed melt is not sufficient to assess model performance.

The dataset and SEB analysis are important for the community in particular for model validation and deserve to be published. However, in its present state, the modelling analysis is not sufficiently robust, because 1) raw data still present biases, 2) models are too different to be compared, 3) none of the models is currently sufficiently accurate to be referred to as the reference. In particular, the authors write that "Bringing the predicted melt rates by these highly parameterized models in agreement with the observed ones seems to be rather a curve adjustment exercise than a indicator of correct physics." However, I feel that this opinion is not supported by results.

I propose authors produce a (real) reference modelling, using adapted calibration for each glacier (see point 2), and then compare this reference with other "simplified" approaches (i.e. the EB-model and the COSIMA model as presented in the present version of the paper). When radiative fluxes are modelled, I propose to perform a calibration/validation step. Finally, in the discussion, I suggest that authors cautiously consider the differences in surface states, elevation and latitude (in particular when the authors compare ablation amounts). For this task, I have a few suggestions which may improve the accuracy of results:

1) A pre-treatment of field data has not been made in depth. Indeed, LW data are biased by very large artefacts, and this introduces large uncertainties in the SEB analysis. + Large biases are observed in LWin and LWout data. These are clearly visible in figure 4. They are likely due to CNR4 heating caused by solar radiation, or to an incorrect calibration. I personally worked during 20 years with these sensors (CNR1, CNR4) and I never observed continuous biases of about 20 W m-2. Obleitner and de Wolde (1999) proposed bias corrections as a function of SWin values, but this does not remove potential biases during the nights. In order to remove biases, I propose that authors analyse LWout values when the surface is melting (LWout should be 315 W m-2). LWnet is possibly correct because corrections may be similar for both LWin and LWout. + Potential other artefacts are not discussed in the text (existing shadows on sensor caused by the station mast, snow accumulation on CNR4, etc). Generally, nonaspirated temperature sensors may be biased high by solar radiation at wind speeds less than 3 m s-1 (Huwald et al., 2009; Georges and Kaser, 2002). A correction for the solar bias is complex but possible. At least, observed temperatures may be flagged when low wind speeds (i.e.,

impact on turbulent heat fluxes calculations. + Data gaps are also not described. =>I suggest that authors accurately correct field data.

2) The "reference model" is not accurate enough due to assumptions made on Ts and turbulent heat flux calculations. SWnet and LWnet would possibly be accurate because these values are directly measured (if biases are removed), but turbulent heat fluxes are clearly not accurate in the reference model, because 1) surface temperature Ts is assumed to be at 0°C, and 2) calculations are done without considering stability conditions in the surface boundary layer. Conversely, the COMISA approach very likely produces more accurate LH and SH, but then the radiation terms are not accurate (see the important differences with observed fluxes in Table A1). =>I propose to run the COMISA model using the measured SWin, Albedo, LWin (after correction), T, Rh and wind speed, solid precipitation, the initial snow height and cloudiness (I am not sure that this variable will impact results since LWin and SWin will be already assimilated). I propose to force the model using specific surface roughness length values according to the surface state. Indeed, Bello Glacier (see photograph) presents small penitents at the surface when Tyndall presents a very smooth surface. Surface roughness length values have already been proposed in the literature for the different studied areas.

3) LW/SW schemes have never been calibrated/validated in the study. When modelled (i.e., in the simplified modelling approaches), albedo and LWin/SWin schemes could be calibrated using a simple monte carlo approach (scores could be computed using field measurements of albedo and LWin/SWin). A sensitivity analysis could also offer interesting information for the discussion. For turbulent heat fluxes, authors could test various surface roughness length values given in the literature. Validation of the "reference model" could also be done between modelled and observed Ts. The modelled ablation could be compared with observation on stakes or on sonic gauges (when available). =>I propose that authors calibrate the different schemes on one period and validate it on another one. This is possible at least for Bello and Tyndall glaciers.

4) How do authors compute melting? Do they use the mean daily energy excess,
or do they compute melting at a 1h time step? This is crucial because calculations must consider that surface melting depends on heat storage in the subsurface. In particular, the existence of subsurface fluxes is never mentioned (heat conduction and transmission of solar radiation in snow and ice) even in the COSIMA model. These fluxes are crucial to explain daily melt intensities. => Does COSIMA compute these fluxes? if not, would it be more accurate to run a model that includes these fluxes? (see for instance Thomas Mölg's model (e.g., Mölg et al., 2008, 2009b, 2012; Gürgiser et al., 2013)). If subsurface fluxes are not considered, please justify this assumption.

5) Finally, a deep review on SEB modelling in the Andes is lacking. Many SEB modelling are available along the Andes, but currently the review of literature is limited to (MacDonnell et al., 2013; Brock et al. 2007; Schneider et al., 2007; Pellicciotti et al., 2008; Ayala et al., 2017; Sicart et al., 2008), which is not up-to-date (papers mainly refer to studies published more than 10 years ago). A more exhaustive review of studies performed along the Andes would offer interesting information for the discussion.

As a summary, I suggest that the authors take a slightly different approach in order to present their study. I suggest to: a) compute a real reference SEB b) describe the differences in fluxes according to the altitude/latitude, c) compute simplified EB-model and COSIMA modelling. d) conclude on the differences existing between the "simplified approaches" and the reference model e) Please reconsider the conclusion of the paper if you don't clearly demonstrate that SEB modelling is harder to apply than an empirical model, and that it does not offer better results. f) I suggest that authors make a thorough editing of the text to improve the language style.

Minor Comments,

Line 5: Please write "Turbulent sensible heat flux", "Turbulent latent heat flux" and "turbulent heat fluxes"

Line 9 : transport coefficient => do you mean bulk exchange coefficient? Or bulk transfer coefficient?
Line 20: "These kind of models are sometimes called "physical melt models" => please include references, at least in the Andes.

Section 1 : Introduction => the introduction is confusing and is not focusing on the main objective of the paper. This introduction should present the interest of SEB modelling on glaciers, and a review of knowledge on the SEB in the Andes and under similar climates. I propose that authors reorganize the introduction and remove several sentences: For instance, the paragraph "Chile is well-known [...] future surface mass balance and melt water discharge of Chilean glaciers" could be included in a subsection of the section "sites" which could be titled "climate settings".

Page 2 Line 1 : "Chile is well-known for the climatic variety due to its north-south extension of the territory" => why do the authors focus on Chile only, and not on Chile/Argentina? please rewrite.

Page 2 Line 2 : "Pacific anticyclone plays a key role" => on what?

Page 2 Line 3: "sub-glaciological zones" => please cite (Braun et al., 2019) and (Dus-saillant et al., 2019)

Page 2 line 14 : Are you sure that the mega-drought reached Patagonia?

Page 2 line 16 : "Chile hosts the majority of glaciers in South America (more than 80% of the area)," => what about Argentina?

Page 2 line 17 : Âń which are mostly thinning and retreating in the last decades (e.g. Braun et al. (2019))." => please also cite (Dussaillant et al., 2019)

Page 2 line 17-20, and elsewhere: "The projections of future changes [...] climatological zones is necessary" => glacier wastage projections for calving glaciers are impossible if we only consider the SMB and the SEB (e.g., Collao-Barrios et al., 2018). The authors never write that Exploradores and Tyndall are calving glaciers. Please comment this point. TCD
Page 2 line 21: "There exist few surface mass balance observation programs on Chilean glaciers:" => why considering only Chile?.

Page 2 line 23: "Echaurren Norte Glacier" => please introduce also the Piloto glacier and other glaciers under study in Argentina.

Page 2 line 29 : "the limited accumulation of snow is not able to make up with the ablation processes" => If a glacier is present, this means that there was a time when the glacier had a positive SMB.

Page 4, line 17: "glacier melt at equator near Zongo Glacier" => Zongo Glacier is in Bolivia, not at the equator.

Page 4, line 26-29: "in this study we want to test their ability to reproduce the individual energy fluxes" => This is relevant, but this is not really done in the paper. I suggest that authors compare each modelled and observed fluxes, with figures and statistics.

"we want to emphasize the differences between the model parameterizations and their ability to reproduce the directly measured radiative fluxes at the glacier surfaces" => It is currently hard to conclude, because the authors use 3 very different models, with different assumptions on many variables (SWin, LWin, albedo, Ts, and turbulent heat fluxes). It would be easier to force COSIMA with observations in order to produce a reference modelling, then make simplified EB-model and COSIMA approaches. Another interesting way to reach conclusions would be to make a sensitivity analysis on COSIMA model.

"We also compare three different parameterization for the turbulent fluxes of sensible and latent heat" => Here, the authors used different equations, in which stability calculations were simplified, but they also changed the values of Ts. It is thus really hard to conclude on comparisons.

Section 2: Sites => please present different regions using a bulleted list and include here the paragraph which is currently in the introduction. Piramide, San Francisco,

TCD
and Bello glaciers should enter in the same region; Tyndall and Exploradores glaciers would be in the of Patagonia. Please remind in the text that Tyndall and Exploradores are calving glaciers.

Section 3: Methods Page 5, Line 16: "to a very good approximation sum up to the melt energy available at the glacier surface" => Here and elsewhere in the text: please include values, scores, statistics.

Page 5, Line 18: Heat conduction and solar radiation transfer in the ice are neglected when they play a crucial role even in summer (See for instance Gürgiser et al., 2013). Please justify the choice of neglecting these fluxes. Are they considered in the COSIMA model?

Section 3.1: could be included in sites. Please inform on potential data gaps, and data treatment. Please remove biases in the data (see point 1).

Section 3.2 Reference database: I suggest that authors change the reference modelling as described in the introduction of my review.

Page 7, line 7 : "The bulk aerodynamic approach is employed" => The bulk aerodynamic approach is not never fully used here. The methods used here are simplified approaches.

Page 7 Line 14 : "T(s) is the temperature of the glacier-atmosphere interface, which is assumed to be  $0^{\circ}C$ "=> Please use corrected LWout (using obleitner and deWolde (1999) approach) to compute Ts.

Equation (2) : is only valid for neutral surface boundary layer and assuming that zOm = zOT = zOq, These points are suggested before in the text (see "the eddy diffusivity for heat has the same value as the eddy diffusivity for water vapor and the eddy viscosity"), but writing that the SBL is assumed to be neutral is more direct.

Page 7, line 22: "constant roughness length of z0=0.5mm" => please discuss this value because turbulent heat fluxes directly depend on it (they are twice larger if z0 is

TCD
10 times larger). z0 is expected to differ between snow and ice, and have very specific values over penitentes. There are many references proposing values assuming that z0 = z0m = z0T = z0q. Another option would be to consider different z0m for snow and for ice, and then apply Andreas [1987] polynomials.

Figure 3a. I don't understand why the authors don't use (Goff, J.A., and Gratch, S. 1946). Relationships. In Particular, the COSIMA relationship looks like erroneous. Moreover, I don't understand why two curves are given above  $0^{\circ}$ C (one for ice, and the other for snow), and only one below  $0^{\circ}$ C, when it should be the reverse (saturation against solid or liquid phase makes sense below  $0^{\circ}$ C).

Figure 3b. The differences are so large that the authors could calibrate a relationship between atmospheric emissivity, T and Rh, using their field data. They also could consider MacDonell et al., (2013) study.

Section 3.3, EB-Model Page8 Line 13: "the sum of the direct and diffuse incoming solar radiation" => Does EB-Model consider data from a DEM to compute Diffuse/direct components? How is it computed in the case of overcast conditions?

Page 9 line 1: It seems strange to assume a constant albedo on a glacier, and snow patches, when solid precipitation occurred. It would be better to consider the albedo scheme from COSIMA.

Page 9, Line 5: LWout= is 315.6 W/m2 => this assumption is really strong again. If you consider that melt is observed when values are maximum, then refreezing is observed at night in Figure 4 (except for Tyndall).

Page 9 line 10: "clear sky emissivity"=> again, validation of the equation may be easily done using field data.

Page 9 Line 12:" theoretically site-specific clear sky incoming solar radiation": how is it computed? Moreover, do the authors compute n with same equation in every models? Is it computed with equation 9?

TCD
Equations 7 and 8 or only for stable conditions, which are generally not verified in the morning. Why do they assume this, when computing a Richardson number is very easy when surface temperature is available? Please justify.

Page 9 line 22: "the roughness Reynolds number (Brock and Arnold, 2000)." => you mean using Andreas (1987) polynomials?

Page 9, Line 31 : "theoretical, site specific clearsky radiation computed by a code developed by Corripio (2003)"=> Do you mean SOLTRAN? Please give the exact reference.

Page 10 Line 6: please cite : U.S. Army Corps of Engineers (1956).

Page 10, Line 22: "However here SH is multiplied by a correction factor which depends on the bulk Richardson number Ri" => Please precise. Why do the authors use this formulae instead of the bulk approach? In particular, the assumptions behind equation 13 are not clear to me. Could they compute the turbulent heat fluxes offline using the bulk method and Ts, Tair, Rh, and U given by COSIMA and compare the results with COSIMA's turbulent heat fluxes?

Section 4.1 : Glacier climate => please reformulate this title Table 3: I suggest that the authors make a bulleted list and first compare glaciers in the same region, and then compare glaciers at different locations. Please discuss the differences in altitude and surface state in a same region.

Page 11, Line 15: "incoming longwave radiation increases from [...]increased cloudiness in the Wet Andes." => elevation and temperature also play a key role here. In particular, differences in LWin between Bello, Piramide and San Francisco are largely related to temperature.

Figure 4f: the surface at Tyndall glacier is constantly melting. This suggests that sensors are biased by a constant value of 15Wm-2. This bias seems to be retrieved on other glaciers (except on exploradores, where it seems to be even larger).
Section 4.2: Page 13, Line 3: "we exclude Pirámide Glacier from" => Instead of removing this glacier, I suggest that authors compare with results on Pichillancahue-Turbio Glacier (Brock et al., 2007).

Page 13, Line 13: "The predicted melt rates are higher for Patagonian Glaciers as compared to the Glacier in the Central Andes" => This sentence does not make sense, melting rates depend on elevation. At 4000 m asl, melting is zero in Patagonia.

Figure 6, 7 and 8: show that the big differences between the 3 models are observed in LWnet and SWnet, probably due to the very strong assumptions made on LWout and albedo variations. But if we analyse Table A1, the main differences are observed in the mean SH and LH values. I suggest that authors discuss this point.

Page 15, Line 10: "although they have opposite exposition" => The studied surfaces are expected to be flat (no slope).

Page 15, Line 16: "LWout detected on the glaciers are surprisingly higher than the expected 315.6 W/m2" => please correct biases in data before analysing the SEB.

Caption of Figure 9: "albedo. In COSIMA the following parameters have been chosen: frsnow=0.8, firn=0.5, time constant t= 2 days, snow depth constant d= 8 cm (see equations (11) and (12)" => How did the authors calibrate these parameters?

Page 17, Line 14 (and Page 18, Line 3): "At Exploradores Glacier the measured emissivity reaches values higher than one."=> Please correct LWin before computing the emissivity.

Page 18, Line 8 (and the end of the paragraph): "The variability of the modeled turbulent fluxes is very similar in all three methods[...] have very similar aspect: in all approaches the sensible heat flux is mainly driven by the temperature difference"=> please refer to Table A1. For instance, if we consider the Bello Glacier, SH is ranging from 6 to 30 W m-2 and LH from -23 to -53 Wm-2. Considering the sum of turbulent heat fluxes, SH+LH is ranging from -40 to +7 W m-2. This maximum difference beTCD
tween the 3 models (47 W m-2) is larger than those observed in LWnet (22 W m-2) and in SWnet (15 W m-2). Perhaps I did not understand the end of the paragraph, but the large difference in turbulent heat fluxes results from the very different assumptions done on Ts and on stability corrections.

Section 5.3: melt rates: I propose that authors include a table to allow a quick validation between modelled ablation and observed ablation from stakes or sonic gauges. In this Table, it would be interesting to include the elevation of ablation measurements, the latitude, the time period of measurements.

Page 18, line 23 :" observed melt range"=> "observed melt rates"

Page 18, line 35: This comparison looks strange because Exploradores is located in campo de Hielo Norte, whereas Grey Glacier is located in Campo de Hielo Sur.

Page 19, Line 23: "The capacity of the models to reproduce the measured radiative fluxes is still improvable." => This conclusion looks strange when only 3 simple approaches are used. Please note that there is a large number of complex snow models and many complex experiments are done to improve these models (e.g., ESM-SnowMIP (Krinner et al., 2018).

Page 19, Line 28 : "This is because the used parameterizations are fits to data that were obtained in different climatic conditions." => This sentence seems trivial. I propose that authors calibrate their model with their observation, and then perform a model sensitivity analysis.

Page 19 Line 31: "Different parameterizations for the turbulent fluxes of sensible and latent heat were compared in this study,"=> because assumptions made on Ts and equations used were different in the 3 approaches, it is hard to conclude.

Page 19 Line 33 (and page 20, Line 1): "There are many parameters involved in these parameterizations"=> I don't agree, the bulk method only requires to define z0m.

Page 20, second paragraph: "Bringing the predicted melt rates by these highly pa-
rameterized models in agreement with the observed ones seems to be rather a curve adjustment exercise than a indicator of correct physics." => Perhaps this conclusion is real for EB-model and COSIMA, but I would not extrapolate this conclusion to all the SEB models.

Page 20, Line 18: "The inferred melt rates were higher for the Patagonian Andes than for the Central Andes."=> it depends on elevation of study sites. Melt is zero in Patagonia at 4000 m asl.

Page 20, Line 20: "The models underestimated the measured emissivity of the clearsky atmosphere in the Wet Andes."=> please correct LWin before concluding.

Page 20, line 23 : "To develop or improve physical models we have to validate every single model parameterization against data"=> this could be done in present study.

References:

Ayala, A., Pellicciotti, F., MacDonell, S., McPhee, J., and Burlando, P.: Patterns of glacier ablation across North-Central Chile: Identifying the limits of empirical melt models under sublimation-favorable conditions, Water Resources Research, 53, 5601–5625, https://doi.org/10.1002/2016WR020126, 2017.

Braun, M.H., Malz, P., Sommer, C. et al. Constraining glacier elevation and mass changes in South America. Nature Clim Change 9, 130–136 (2019) doi:10.1038/s41558-018-0375-7

Brock, B., Rivera, A., Casassa, G., Bown, F., and Acuna, C.: The surface energy balance of an active ice-covered volcano: Villarrica Volcano, Southern Chile, in: Annals of Glaciology, VOL 45, 2007, edited by Clarke, G. and Smellie, J., vol. 45 of Annals of Glaciology-Series, pp. 104–114, https://doi.org/10.3189/172756407782282372, international Symposium on Earth and Planetary Ice-Volcano Interactions, Univ Iceland, Inst Earth Sci, Reykjavik, ICELAND, JUN 19-23, 2006, 2007.

Collao-Barrios, G., F. Gillet-Chaulet, V. Favier, G. Casassa, E. Berthier, I. Dussaillant, P.

TCD
Gourlet, J. Mouginot, E. Rignot and M. Schaefer (2018) Ice-flow modelling to constrain the surface mass balance and ice discharge of San Rafael Glacier, Northern Patagonia Icefield, Journal of Glaciology, 1-15, https://doi.org/10.1017/jog.2018.46.

Corripio, J.: Vectorial algebra algorithms for calculating terrain parameters from DEMs and solar radiation modelling in mountainous terrain, International Journal of Geographical Information Science, 17, 1–23, https://doi.org/10.1080/713811744, 2003.

Dussaillant, I., Berthier, E., Brun, F. et al. Two decades of glacier mass loss along the Andes. Nat. Geosci. 12, 802–808 (2019) doi:10.1038/s41561-019-0432-5

Georges, C., and G. Kaser (2002), Ventilated and unventilated air temperature measurements for glacier? Climate studies on a tropical high mountain site, J. Geophys. Res., 107(D24), 4775, doi:10.1029/2002JD002503.

Goff, J.A., and Gratch, S. 1946. Low-pressure properties of water from -160 to 212 °F. In Transactions of the American Society of Heating and Ventilating Engineers, pp 95–122, presented at the 52nd annual meeting of the American Society of Heating and Ventilating Engineers, New York, 1946.

Gurgiser, W., Marzeion, B., Nicholson, L., Ortner, M., and Kaser, G.: Modeling energy and mass balance of Shallap Glacier, Peru, The Cryosphere, 7, 1787–1802, https://doi.org/10.5194/tc-7-1787-2013, 2013.

Huwald, H., C. W. Higgins, M.-O. Boldi, E. Bou-Zeid, M. Lehning, and M. B. Parlange (2009), Albedo effect on radiative errors in air temperature measurements, Water Resour. Res., 45, W08431, doi:10.1029/2008WR007600.

Krinner, G., Derksen, C., Essery, R., Flanner, M., Hagemann, S., Clark, M., Hall, A., Rott, H., Brutel-Vuilmet, C., Kim, H., Ménard, C. B., Mudryk, L., Thackeray, C., Wang, L., Arduini, G., Balsamo, G., Bartlett, P., Boike, J., Boone, A., Chéruy, F., Colin, J., Cuntz, M., Dai, Y., Decharme, B., Derry, J., Ducharne, A., Dutra, E., Fang, X., Fierz, C., Ghattas, J., Gusev, Y., Haverd, V., Kontu, A., Lafaysse, M., Law, R., Lawrence, TCD
D., Li, W., Marke, T., Marks, D., Ménégoz, M., Nasonova, O., Nitta, T., Niwano, M., Pomeroy, J., Raleigh, M. S., Schaedler, G., Semenov, V., Smirnova, T. G., Stacke, T., Strasser, U., Svenson, S., Turkov, D., Wang, T., Wever, N., Yuan, H., Zhou, W., and Zhu, D.: ESM-SnowMIP: assessing snow models and quantifying snow-related climate feedbacks, Geosci. Model Dev., 11, 5027–5049, https://doi.org/10.5194/gmd-11-5027-2018, 2018.

MacDonell, S., Kinnard, C., Mölg, T., Nicholson, L., and Abermann, J.: Meteorological drivers of ablation processes on a cold glacier in the semi-arid Andes of Chile, The Cryosphere, 7, 1513, 2013a.

MacDonell, S., L. Nicholson, and C. Kinnard (2013b), Parameterization of incoming longwave radiation over glacier surfaces in the semiarid Andes of Chile, Theor. Appl. Climatol., 111, 513–528, doi:10.1007/s00704-012-0675-1.

Mölg, T., Cullen, N. J., Hardy, D. R., Kaser, G., and Klok, L.: Mass balance of a slope glacier on Kilimanjaro and its sensitivity to climate, Int. J. Climatol., 892, 881–892, doi:10.1002/joc.1589, 2008.

Mölg, T., Cullen, N. J., and Kaser, G.: Solar radiation, cloudiness and longwave radiation over low-latitude glaciers: implications for mass-balance modelling, J. Glaciol., 55, 292–302, doi:10.3189/002214309788608822, 2009b.

Mölg, T., Maussion, F., Yang, W., and Scherer, D.: The footprint of Asian monsoon dynamics in the mass and energy balance of a Tibetan glacier, The Cryosphere, 6, 1445–1461, doi:10.5194/tc-6-1445-2012, 2012.

Obleitner, F., and J. de Wolde (1999), On intercomparison of instruments used within the VatnajoÂĺkull glacio-meteorological experiment, Boundary Layer Meteorol., 92, 27–37.

Pellicciotti, F., Helbing, J., Rivera, A., Favier, V., Corripio, J., Araos, J., Sicart, J.-E., and Carenzo, M.: A study of the energy balance and melt regime on Juncal Norte Glacier,
semi-arid Andes of central Chile, using melt models of different complexity, HYDRO-LOGICAL PROCESSES, 22, 3980–3997, https://doi.org/10.1002/hyp.7085, workshop on Glaciers in Watershed and Global Hydrology, Obergurgl, AUSTRIA, AUG 27-31, 2007, 2008.

Schneider, C., Kilian, R., and Glaser, M.: Energy balance in the ablation zone during the summer season at the Gran Campo Nevado Ice Cap in the Southern Andes, Global and Planetary Change, 59, 175 – 188, https://doi.org/http://dx.doi.org/10.1016/j.gloplacha.2006.11.033, http://www.sciencedirect.com/science/article/pii/S092181810600292X, mass Balance of Andean Glaciers, 2007.

Sicart, J. E., Hock, R., and Six, D.: Glacier melt, air temperature, and energy balance in different climates: The Bolivian Tropics, the French Alps, and northern Sweden, Journal of Geophysical Research: Atmospheres, 113, 2008.

U.S. Army Corps of Engineers (1956), Summary report of the snow investigations, snow hydrology, North Pac. Div., Portland, Oreg.

---

## Author Comment (AC1) · 20 Dec 2019

Comments for the Authors:

The paper addresses relevant scientific questions within the scope of TC. It shows a good data set of meteorological measurements and estimates of the surface energy balance of Chilean glaciers of contrasting climates.
Unfortunately, in my opinion the manuscript does not represent substantial progress beyond current scientific understanding. The overall presentation is well structured and clear. However, the text can be improved; it should be more concise and accurate. I found that some scientific methods were not suitable and that some assumptions were not clearly outlined. Finally, the discussion does not reach substantial conclusions. In my opinion, the main problem is that the objectives of the study are not clearly defined. The manuscript presents various results but the overall purpose of the study remains imprecise. The analysis of this interesting dataset could be an interesting contribution to the literature if the objectives were articulated more clearly and reflected in an appropriate methodology.

The paper would certainly need major improvements to be innovative and merit publication in TC. These points are detailed in the comments below.

Major comments:
The objectives should be redefined. The main contribution of the study is the presentation of a comprehensive set of meteorological measurements on glaciers of contrasting climates along the Andes. The data certainly allow for correct estimation of the energy fluxes at the surface of the glaciers. Radiation flux measurements seem appropriate and accurate. Turbulent flows are not measured directly, but meteorological measurements probably allow correct estimates(if the appropriate methods are applied, see comments below). Thus, I would suggest focusing on a comparison of the energy fluxes partitioning in the different climates from the dry Andes to the wet Andes (the measurements above debris-covered ice are not useful here). The effects of latitude and of altitude on the energy fluxes need to be discussed in more detail.
**Ok. Section 4.1 was revised completely.**
One limitation of the study is that general interpretations of different climates are deduced from point-scale measurements. The partitioning of the energy fluxes depends on the position of the weather station. For instance, the albedo varies greatly over short distances near the snow line, so that the interpretation of punctual energy flux measurements can lead to an erroneous generalization of the melting characteristics to the entire ablation area. This point should be discussed.
**Ok. This point is mentioned now.**

Some applied methods are not suitable The NR-Lite sensor is less accurate than the CNR4. Thus, calculations of long wave radiation fluxes can be problematic on the Mocho glacier. **Ok. We agree that it is difficult to draw sound conclusions about longwave radiative fluxes using the data from NR-Lite sensor. Still, we think that the results for the longwave net fluxes obtained on Mocho fit well in the general North-South trend, which makes us more confident about them.**

Comparing different models does not bring much newness here. Some assumptions are not valid (Ts=0∘C, constant albedo...) or some formulations are not adequately described (see below).

I would suggest discussing the energy fluxes derived from the most direct approach: the 'reference database' based on measurements. For example,there is no need to assume that the surface temperature is fixed to 0∘C (P7, line 14)if outgoing longwave measurements are available. This assumption (which does not seem valid on the San Francisco and on Bello glaciers) has a significant impact on the turbulent sensible heat fluxes derived from the bulk method.

**Ok. For the computation of the Reference Database, we compute now the surface temperature using bias-corrected outgoing longwave radiation data. Comparing the performance of the different model parameterizations under different climatic conditions is the core part of this study.**

P8, Equation 4: use the standard relationship of saturation vapor pressure as a function of temperature, no need to test different parametrizations.
**It seems there exist several "standard" parameterizations in the literature. We used the one which we think interpolated best the measurements.**
P8, line 13: why mentioning direct and diffuse components of solar irradiance if global radiation is directly measured?
**Here we are describing the parameterizations of one of the models which we are applying. In this model the division into direct and diffuse components is used to infer the cloud cover.**
Sections 3.2, 3.3 and 3.4: the comparison of the different turbulent 'transfer coefficient'(P7, line 16) should refer to stability concepts. A stability correction must be included over glacier surfaces, using the Monin-Obukhov length scale or the Richardson number. This important point should be clarified. The values of the roughness lengths for momentum, temperature and humidity should also be discussed in more detail with reference to the state of art.
**Ok we added some discussion on that.**

P11, lines 9-12: no need to compare the two methods (especially if they give 'similar results').
**We think that it makes our results more robust, when two different ways of computing albedo give similar results.**

The effects of cloud cover in the different climates should be investigated more rigorously. The applied method is inappropriate (the text does not say how the cloud cover is calculated) and appropriate references are missing. Figure 10 is unclear and its interpretation P17-18 remains vague. Robust methods have been proposed for estimating cloud cover from solar and longwave radiation fluxes measurements (e.g., Marty etal., TAC, 2002 in the Alps; Sicart et al., JOG, 2010 in the tropical Andes; McDonell etal., TAC, 2013 in the semiarid Andes of Chile; Munneke et al., IJC, 2011...). These parametrizations, once calibrated at each site, will make it possible to distinguish clear skies from cloudy conditions. This point, with an adequate methodology, could be an interesting contribution of the study.

**For EB-Model and COSIMA the cloud-cover is estimated using incoming solar radiation fluxes measurements (see section 3.3 and 3.4). Figure 10 has been divided into 4 subplots now, to facilitate the visualization of our analysis.**

Many results are presented without proper interpretation Figures 5, 6, 7 and 8 show many results, but most of them remain poorly analysed. The turbulent fluxes are not measured directly, so their estimates are not very accurate. The large differences in sensible heat flux in the different climates are certainly significant. However, I think no much can be said about the latent heat fluxes shown in figure 5; the fluxes remain close to zero and the uncertainties are certainly large.

**We do not agree here with your statement about the latent heat fluxes: these fluxes depend on the moisture content of the atmospheric layer next to the glacier surface. If the moisture content in this layer is less than the moisture content of a saturated air layer at zero degrees Celsius just above the glacier surface, then the latent heat flux has a negative sign (moisture is transported from the glacier surface to the atmosphere, which causes more evaporation or sublimation to happen on the glacier surface, which are processes**
**that consume energy). This is happening in the Central Andes. If on the other hand the moisture content of the lowest atmospheric layer is higher than the moisture content of a saturated air layer at zero degrees Celsius just above the glacier surface, then the latent heat flux has a positive sign (moisture is transported from the atmosphere to the glacier surface, which causes condensation to happen on the glacier surface, which is processes that provides energy). This is happening in the Patagonian Andes**
**( especially Exploradores Glacier).**

- The text must be carefully proofread. Be more specific and accurate, for instances:

- P2, lines 23-25: give numbers to quantify the trends in mass balance

**ok, number was added.**

- P5: the correct terms are 'irradiance' or 'radiationfluxes' (in W/m2)

**The correct terms for what on page5? Not all energy fluxes on the glacier are radiative!**

- P13, line 6 'ice equivalent'? Do you mean water equivalent?

**As the glaciers surfaces to which we applied the model approach were mostly ice, we decided to present the results in 'ice equivalent'**

- P13 where are the figures 12 and 13?

**Sorry, these figures existed in an earlier version of the manuscript. They were removed now!**

- P16, line 16: Equation 12?

**Sorry, we meant equation (11)**

- Where is Table 4?

**Sorry. We refer to table A 1. Changed!**

Minor comments:

- P16, lines 2-4: the effects on longwave radiation of the "very humid and temperate air column between the sensor and the glacier surface" is probably small and can be estimated [e.g., Pluss and Ohmura, 1997].

**Ok, now we bias-corrected the measured longwave radiation following a suggestion form reviewer2.**

- P19: why not using the meteorological and ablation measurements on the Bello glacier during the summers 2013/14 and 2014/2015 to validate (in a rigorous way)the calculations of the energy fluxes?

**We are using data from summer 2014/2015, since in summer 2013/2014 there was a problem with the sensor which is measuring shortwave radiation. Sadly for the summer 2014/2015 no ablation measurements are available at the location of the automatic weather station (see page 87 in CEAZA: Modelación del balance de masa y descarga de agua en glaciares del Norte Chico y Chile Central, Tech. rep., Dirección General de Aguas, S.I.T. No. 382, 2015.)**

---

## Author Comment (AC2) · 20 Dec 2019

This paper focuses on calculations of the surface energy balance (SEB) of glaciers along the Andes of Chile. The analysis is covering a large latitudinal range, between18∘S and 55∘S, and tries to describe the main differences existing in the processes controlling melting under diverse climate settings. The paper compares three differentmodelling approaches applied on a dataset of 6 glaciers. The authors intend determining adequate parameterization of SEB models and conclude that the use of observed melt is not sufficient to assess model performance.The dataset and SEB analysis are important for the community in particular for model validation and deserve to be published. However, in its present state, the modelling analysis is not sufficiently robust, because
1) raw data still present biases,

2) models are too different to be compared,

3) none of the models is currently sufficiently accurate to be referred to as the reference.

In particular, the authors write that "Bringing the predicted melt rates by these highly parameterized models in agreement with the ob-served ones seems to be rather a curve adjustment exercise than a indicator of correct physics.". However, I feel that this opinion is not supported by results.I propose authors produce a (real) reference modelling, using adapted calibration for each glacier (see point 2), and then compare this reference with other "simplified" approaches (i.e. the EB-model and the COSIMA model as presented in the present version of the paper). When radiative fluxes are modelled, I propose to perform acalibration/validation step. Finally, in the discussion, I suggest that authors cautiously consider the differences in surface states, elevation and latitude (in particular when the authors compare ablation amounts). For this task, I have a few suggestions which may improve the accuracy of results:

1) A pretreatment of field data has not been made in depth. Indeed, LW data are biased by very large artefacts, and this introduces large uncertainties in the SEB analysis. + Large biases are observed in LWin and LWout data. These are clearly visible in figure 4. They are likely due to CNR4 heating caused by solar radiation, or to an incorrect calibration. I personally worked during 20 years with these sensors (CNR1,CNR4) and I never observed continuous biases of about 20 W m-2. Obleitner and deWolde (1999) proposed bias corrections as a function of SWin values, but this does not remove potential biases during the nights. In order to remove biases, I propose that authors analyse LWout values when the surface is melting (LWout should be 315 W m-2). LWnet is possibly correct because corrections may be similar for both LWin and LWout.+ Potential other artefacts are not discussed in the text (existing shadows on sensor caused by the station mast, snow accumulation on CNR4, etc). Generally, non aspirated temperature sensors may be biased

high by solar radiation at wind speeds less than 3 m s−1 (Huwald et al., 2009; Georges and Kaser, 2002). A correction for the solar bias is complex but possible. At least, observed temperatures may be flagged when low wind speeds (i.e., <3 m s−1) are observed during the daytime. This has an impact on turbulent heat fluxes calculations. + Data gaps are also not described. =>I suggest that authors accurately correct field data.

**Ok. We bias-corrected longwave radiative fluxes now, using the assumption that the glacier surfaces are melting ( are at zero degrees Celsius) during afternoon (1 p.m. to 6 p.m.). Data gap treatment is described in the section 3.1.**

2) The "reference model" is not accurate enough due to assumptions made on Ts and turbulent heat flux calculations. SWnet and LWnet would possibly be accurate because these values are directly measured (if biases are removed), but turbulent heat fluxes are clearly not accurate in the reference model, because 1) surface temperatureTs is assumed to be at 0∘C, and 2) calculations are done without considering stability conditions in the surface boundary layer. Conversely, the COMISA approach very likely produces more accurate LH and SH, but then the radiation terms are not accurate (see the important differences with observed fluxes in Table A1). =>I propose to run the COMISA model using the measured SWin, Albedo, LWin (after correction), T, Rh and wind speed, solid precipitation, the initial snow height and cloudiness (I am not sure that this variable will impact results since LWin and SWin will be already assimilated).I propose to force the model using specific surface roughness length values according to the surface state. Indeed, Bello Glacier (see photograph) presents small penitents at the surface when Tyndall presents a very smooth surface. Surface roughness length values have already been proposed in the literature for the different studied areas.

**In the revised manuscript we do not assume Ts to be at 0∘C any more. We compute Ts now using the bias corrected values of LWout (except for Mocho, where we do not have this information). We now use the stability correction implemented in COSIMA for the turbulent fluxes for the reference database. The influence of different surface roughness values is discussed now.**

3) LW/SW schemes have never been calibrated/validated in the study. When modelled(i.e., in the simplified modelling approaches), albedo and LWin/SWin schemes could be calibrated using a simple monte carlo approach (scores could be computed using field measurements of albedo and LWin/SWin). A sensitivity analysis could also offer interesting information for the discussion.

**SWin are input data for the modeling approaches. SWout, that is parameterization for albedo, are calibrated and validated on Mocho Glacier (Figure9). LWin is validated in Figure10.**

For turbulent heat fluxes, authors could test various surface roughness length values given in the literature. Validation of the"reference model" could also be done between modelled and observed Ts. The modelled ablation could be compared with observation on stakes or on sonic gauges (when available). =>I propose that authors calibrate the different schemes on one period and validate it on another one. This is possible at least for Bello and Tyndall glaciers.

**Ok, now we discussed the influence of the roughness length on our results. Sadly no direct ablation measurements are available for the modeling periods.**

4) How do authors compute melting? Do they use the mean daily energy excess, or do they compute melting at a 1h time step? This is crucial because calculations must consider that surface melting depends on heat storage in the subsurface. In particular, the existence of subsurface fluxes is never mentioned (heat conduction and transmission of solar radiation in snow and ice) even in the COSIMA model. These fluxes are crucial to explain daily melt intensities. => Does COSIMA compute these fluxes? if not, would it be more accurate to run a model that includes these fluxes?(see for instance Thomas Mölg's model (e.g., Mölg et al., 2008, 2009b, 2012; Gürgiseret al., 2013)). If subsurface fluxes are not considered, please justify this assumption.

**Subsurface fluxes are modeled by COSIMA but are not considered in the other approaches. Considering that we are modelling summer period, the temperature gradients in the snow/ice should be small and the corresponding subsurface fluxes as well ( see calculations below).**

5) Finally, a deep review on SEB modelling in the Andes is lacking. Many SEB modelling are available along the Andes, but currently the review of literature is limited to(MacDonnell et al., 2013 ; Brock et al. 2007; Schneider et al., 2007 ; Pellicciotti et al.,2008 ; Ayala et al., 2017; Sicart et al., 2008), which is not up-to-date (papers mainly re-fer to studies published more than 10 years ago). A more exhaustive review of studies performed along the Andes would offer interesting information for the discussion.

**Ok. Since this work is about Chilean Glaciers we restricted the literature review mainly to studies on Chilean Glaciers. We are happy to receive additional recommendations for studies of relevance for our work to include them in the literature review.**

As a summary, I suggest that the authors take a slightly different approach in order too present their study. I suggest to: a) compute a real reference SEB b) describe the differences in fluxes according to the altitude/latitude, c) compute simplified EB-model and COSIMA modelling. d) conclude on the differences existing between the"simplified approaches" and the reference model e) Please reconsider the conclusion of the paper if you don't clearly

demonstrate that SEB modelling is harder to apply than an empirical model, and that it does not offer better results. f) I suggest that authors make a thorough editing of the text to improve the language style.

**Thank you for your constructive comments. We think that our re-submission is quite in the line of what you proposed.**

Minor Comments,
Line 5: Please write "Turbulent sensible heat flux", "Turbulent latent heat flux" and"turbulent heat fluxes" **ok**
Line 9 : transport coefficient => do you mean bulk exchange coefficient? Or bulktransfer coefficient? **Bulk transfer coefficient (changed)**
Line 20: "These kind of models are sometimes called "physical melt models" => please include references, at least in the Andes. **Ok added.**

Section 1 : Introduction => the introduction is confusing and is not focusing on the main objective of the paper. This introduction should present the interest of SEB modelling on glaciers, and a review of knowledge on the SEB in the Andes and under similar climates. I propose that authors reorganize the introduction and remove several sentences: For instance, the paragraph "Chile is well-known [...] future surface mass balance and melt water discharge of Chilean glaciers" could be included in a subsection of the section "sites" which could be titled "climate settings". **Ok.**

Page 2 Line 1 : "Chile is well-known for the climatic variety due to its north-southextension of the territory" => why do the authors focus on Chile only, and not onChile/Argentina? please rewrite. **We only analyze data from Chilean Glaciers in this manuscript, therefore we think that it is consistent to talk about the climate setting in Chile.**

Page 2 Line 2 : "Pacific anticyclone plays a key role" => on what?
**On the climate in general (formulation changed) .**
Page 2 Line 3: "sub-glaciological zones " => please cite (Braun et al., 2019) and (Dus-saillant et al., 2019) **ok**
Page 2 line 14 : Are you sure that the mega-drought reached Patagonia?
**We are not sure if this drought in Patagonia in 2016 can be associated to the Central Chile mega-drought. We not associate the two phenomena in our manuscript..**
Page 2 line 16 : "Chile hosts the majority of glaciers in South America (more than 80% of the area)," => what about Argentina?
**There is much less glacier area in Argentina, again: our study focuses on Chilean glaciers.**

Page 2 line 17 : ´ n which are mostly thinning and retreating in the last decades (e.g.Braun et al. (2019))." => please also cite (Dussaillant et al., 2019) **ok**

Page 2 line 17-20, and elsewhere: "The projections of future changes [...] climato-logical zones is necessary" => glacier wastage projections for calving glaciers are impossible if we only consider the SMB and the SEB (e.g., Collao-Barrios et al., 2018).The authors never write that Exploradores and Tyndall are calving glaciers. Please comment this point.

**Ok, now we mentioned in section 2 that Exploradores and Tyndall glaciers are calving glaciers.**

Page 2 line 21: "There exist few surface mass balance observation programs on Chilean glaciers:" => why considering only Chile?.

**Because our study is about Chilean Glaciers!**

Page 2 line 23: "Echaurren Norte Glacier" => please introduce also the Piloto glacier and other glaciers under study in Argentina.

**As mentioned above: since this study is about Chilean Glacier, we think it is more consistent to focus on Chilean glaciers in the intro. Again: suggestion of articles about Argentinean glaciers which you estimate of high relevance for our study are very welcome.**

Page 2 line 29 : "the limited accumulation of snow is not able to make up with theablation processes" => If a glacier is present, this means that there was a time when the glacier had a positive SMB.

**ok, we added: "During the monitoring period"**

Page 4, line 17: "glacier melt at equator near Zongo Glacier" => Zongo Glacier is inBolivia, not at the equator.

**Ok, we replaced "equator near" by "tropical"**

Page 4, line 26-29: "in this study we want to test their ability to reproduce the individual energy fluxes" => This is relevant, but this is not really done in the paper. I suggest that authors compare each modelled and observed fluxes, with figures and statistics.

**Modeled and observed fluxes are compared in Figures 5,6,7,8 and model parametrizations are compared to measurements in Figures 9 and 10. The only statistics that are computed up to now are overall mean values and daily mean values. Regarding the great similarity of the course of the melt rates observed in the Figures 6,7,8 we do not think that additional statistics (like correlations and standard deviations) would add some crucial new insights.**

"we want to emphasize the differences between the model parameterizations and their ability to reproduce the directly measured radiative fluxes at the glacier surfaces" =>It is currently hard to conclude, because the authors use 3 very different models, with different assumptions on many variables (SWin, LWin, albedo, Ts, and turbulent heat fluxes). It would be easier to

force COSIMA with observations in order to produce a reference modelling, then make simplified EB-model and COSIMA approaches. An-other interesting way to reach conclusions would be to make a sensitivity analysis on COSIMA model.

**We have chosen a very similar approach now: the reference database is composed of the measured radiative fluxes and turbulent fluxes based on measured surface temperature and using the stability correction which is implemented in COSIMA.**

"We also compare three different parameterization for the turbulent fluxes of sensible and latent heat" => Here, the authors used different equations, in which stability calculations were simplified, but they also changed the values of Ts. It is thus really hard to conclude on comparisons.

**Now the stability correction in the reference database and COSIMA are the same.**

Section 2: Sites => please present different regions using a bulleted list and include here the paragraph which is currently in the introduction. Piramide, San Francisco and Bello glaciers should enter in the same region; Tyndall and Exploradores glaciers would be in the of Patagonia.

**Ok!**

Please remind in the text that Tyndall and Exploradores are calving glaciers.

**Ok, we added that Tyndall and Exploradores glaciers experience some calving as well.**

Section 3: Methods Page 5, Line 16: "to a very good approximation sum up to the melt energy available at the glacier surface" => Here and elsewhere in the text: please include values, scores, statistics.

**Ok, we indicate the maximum difference between our simplified approach and the full COSIMA model now at the end of section 3.4 now.**

Page 5, Line 18: Heat conduction and solar radiation transfer in the ice are neglected when they play a crucial role even in summer (See for instance Gürgiser et al., 2013).Please justify the choice of neglecting these fluxes. Are they considered in the COSIMAmodel?

**Heat conduction through a solid is determined by two things: the thermal conductivity of the solid and temperature gradient. The thermal conductivity of ice is 2.1 W/(mK). The temperature gradient should be zero for the temperate glaciers of Patagonia where mean annual air temperatures are positive. This is also valid for the place on Mocho Glacier, where the energy balance is measured (Schaefer et al. 2017).**

**For the Glaciers in the Central Andes the temperature gradients in the ice should be small by January. Assuming an residual temperature gradient of 1 K/m the heat conduction into the ice would be 2.1 W/m$^2$, a very small value in comparison to the other fluxes.**

**Yes in COSIMA heat conduction is modeled, but for a more consistent comparison of the models we do not consider this fluxes in our study.**

Section 3.1: could be included in sites. Please inform on potential data gaps, and data treatment. Please remove biases in the data (see point 1).

**Data gaps and data treatment are commented. Bias was removed from the longwave radiation data.**

Section 3.2 Reference database: I suggest that authors change the reference modelling as described in the introduction of my review.
**Ok!**

Page 7, line 7 : "The bulk aerodynamic approach is employed" => The bulk aerodynamic approach is not never fully used here. The methods used here are simplified approaches.
**What do yo mean with "fully" used? To our understanding the bulk aerodynamic approach is a way of quantifying the turbulent fluxes by making three important assumption (which are stated in the manuscript). We are happy to get feedback from you in the case you are thinking that we forget to mention another important assumption of this approach.**

Page 7 Line 14 : "T(s) is the temperature of the glacier-atmosphere interface, which is assumed to be 0∘C"=> Please use corrected LWout (using obleitner and deWolde(1999) approach) to compute Ts.
**Ok.**

Equation (2) : is only valid for neutral surface boundary layer and assuming that z0m =z0T = z0q, These points are suggested before in the text (see "the eddy diffusivity for heat has the same value as the eddy diffusivity for water vapor and the eddy viscosity"),but writing that the SBL is assumed to be neutral is more direct.
**Ok we now added an indication to this assumption that it is associated to a neutral atmosphere**

Page 7, line 22: "constant roughness length of z0=0.5mm" => please discuss this value because turbulent heat fluxes directly depend on it (they are twice larger if z0 is 10 times larger). z0 is expected to differ between snow and ice, and have very specific values over penitentes. There are many references proposing values assuming that z0= z0m = z0T = z0q. Another option would be to consider different z0m for snow and for ice, and then apply Andreas [1987] polynomials.
**As indicated in the text, the value 0.5mm, which we chose for z_0 is an intermediate value which is in the range of recommended values for both smooth ice and snow surfaces (page 155, Cuffey&Paterson).**

Figure 3a. I don't understand why the authors don't use (Goff, J.A., and Gratch, S.1946). Relationships.

**Bolton(1980) seems to interpolate the measurements very well.**

In Particular, the COSIMA relationship looks like erroneous.

**You are right. This formula was implemented in the first version of COSIMA that we downloaded and was indicated in Huintjes et al. 2015a. But this seemed to be a bug which was changed now.**

Moreover, I don't understand why two curves are given above 0∘C (one for ice, and the other for snow), and only one below 0∘C, when it should be the reverse (saturation against solid or liquid phase makes sense below 0∘C).

**All the curves are indicated below zero degrees as well, but the different parametrizations overlap for these temperatures, which is the reason why the three curves are not so easy to distinguish**

Figure 3b. The differences are so large that the authors could calibrate a relationship between atmospheric emissivity, T and Rh, using their field data. They also could consider MacDonell et al., (2013) study. Section 3.3, EB-Model

**Sorry, here we used an older version of the Graph where there was a problem with the units of P_vap to calculate the clearsky emissivity. The correct parametrization is shown now in the new graph and the different parametrizations are more similar now. The idea of this piece of work is not to derive new parametrizations but to test transferibility of parametrizations obtained at other glaciers.**

Page8 Line 13: "the sum of the direct and diffuse incoming solar radiation" => Does EB-Model consider data from a DEM to compute Diffuse/direct components?

**No, it considers only reflection from the surroundings in case of the installation of the AWS on a non-zero slope, where part of the slope should be visible for the sensor.**

How is it computed in the case of overcast conditions?

**Increased cloudiness increases the relative contribution of the diffuse radiation ( see formulas (2),(5) and (6) in Brock and Arnold 2000).**

Page 9 line 1: It seems strange to assume a constant albedo on a glacier, and snow patches, when solid precipitation occurred. It would be better to consider the albedo scheme from COSIMA.

**The idea of this study is to test and compare the performance of different parametrizations. The COSIMA albedo scheme is validates against measurements on Mocho Glacier (Figure 9).**

Page 9, Line 5: LWout= is 315.6 W/m2 => this assumption is really strong again. If you consider that melt is observed when values are maximum, then refreezing is observed at night in Figure 4 (except for Tyndall).

**We agree. However this assumption is part of this model that we decided to test in this piece of work.**

Page 9 line 10: "clear sky emissivity"=> again, validation of the equation may be easily done using field data.

**Correct. This is what we are doing in section 5.2 (Figure 10)**

Page 9 Line 12:" theoretically site-specific clear sky incoming solar radiation": how is it computed?

**It is computed according to the formulas derived in Corripio, J.: Vectorial algebra algorithms for calculating terrain parameters from DEMs and solar radiation modelling in mountainous terrain,International Journal of Geographical Information Science, 17, 1–23,**

Moreover, do the authors compute n with same equation in every models?Is it computed with equation 9?

**Equation 9 is only used in the third method, EB-model has its own scheme and for method1 it is not necessary to calculate cloudiness, since the longwave radiation is directly measured.**

Equations 7 and 8 or only for stable conditions, which are generally not verified in the morning.

**During the measurements period air temperature is > 0 degrees Celsius also in the early morning, which should guaranty stability.**

Why do they assume this, when computing a Richardson number is very easy when surface temperature is available? Please justify.

**Again, in this study the primary goal is not to change the different assumptions made by the models employed but to show how these parametrizations vary between the model and what are consequences of this assumptions on the results.**

Page 9 line 22: "the roughness Reynolds number (Brock and Arnold, 2000)." => you mean using Andreas (1987) polynomials?

**Yes, reference added!**

Page 9, Line 31 : "theoretical, site specific clearsky radiation computed by a code developed by Corripio (2003)"=> Do you mean SOLTRAN? Please give the exact reference.

**A collection of codes are used in which the formula derived in Corripio (2003) are applied. These codes are written in IDL.**

Page 10 Line 6: please cite : U.S. Army Corps of Engineers (1956).
**We do not have access to this piece of work!**

Page 10, Line 22: "However here SH is multiplied by a correction factor which depends on the bulk Richardson number Ri" => Please precise.
**Ok. reformulated.**
Why do the authors use this formulae instead of the bulk approach?
**Again, here we use the parametrizations that are already implemented in the models.**
In particular, the assumptions behind equation13 are not clear to me. Could they compute the turbulent heat fluxes offline using the bulk method and Ts, Tair, Rh, and U given by COSIMA and compare the results with COSIMA's turbulent heat fluxes?
**Here it would be good to know what you call THE bulk method. To our understanding (and according to the authors of COSIMA), COSIMA IS using the bulk method. Perhaps it is using different stability corrections to the ones you are used to?**

Section 4.1 : Glacier climate => please reformulate this title
**Ok. Now we call this section: "Microclimatic conditions on the glaciers surfaces"**

Table 3: I suggest that the authors make a bulleted list and first compare glaciers in the same region, and then compare glaciers at different locations. Please discuss the differences in altitude and surface state in a same region.
**Ok. The formulation of this complete paragraph has been changed now following your suggestions.**
Page 11, Line 15: "incoming longwave radiation increases from increased cloudiness in the Wet Andes." => elevation and temperature also play a key role here. In particular, differences in LWin between Bello, Piramide and San Francisco are largely related to temperature.
**Ok, we mention this now.**

Figure 4f: the surface at Tyndall glacier is constantly melting. This suggests that sensors are biased by a constant value of 15Wm-2. This bias seems to be retrieved on other glaciers (except on exploradores, where it seems to be even larger).
**Ok, we removed the bias now!**
Section 4.2: Page 13, Line 3: "we exclude Pirámide Glacier from" => Instead of removing this glacier, I suggest that authors compare with results on Pichillancahue-TurbioGlacier (Brock et al., 2007).
**We do not have measured ablation data for Pirámide Glacier for the study period, which makes it impossible to compare with the results of Brock et al. 2007.**

Page 13, Line 13: "The predicted melt rates are higher for Patagonian Glaciers as compared to the Glacier in the Central Andes" => This sentence does not make sense,melting rates depend on elevation. At 4000 m asl, melting is zero in Patagonia.
**This is right, but this study is about comparing melt rates in the ablation area of the glaciers. We precised this in the text now.**

Figure 6, 7 and 8: show that the big differences between the 3 models are observed inLWnet and SWnet, probably due to the very strong assumptions made on LWout and albedo variations. But if we analyse Table A1, the main differences are observed in the mean SH and LH values. I suggest that authors discuss this point.
**A stability correction of the turbulent fluxes was applied now to in the reference database and the resulting fluxes are very similar to the ones predicted by COSIMA now. Differences between the fluxes  obtained by the different models are discussed in section 5.2.**

Page 15, Line 10: "although they have opposite exposition" => The studied surfaces are expected to be flat (no slope).
**Yes, but the exposition of the glacier still can make a difference due to shading of the the high peaks which constitute the accumulation area of the glaciers.**

Page 15, Line 16: "LWout detected on the glaciers are surprisingly higher than the expected 315.6 W/m2" => please correct biases in data before analysing the SEB.
**Ok. We performed the bias correction now!**
Caption of Figure 9: "albedo. In COSIMA the following parameters have been cho-sen: frsnow=0.8, firn=0.5, time constant t= 2 days, snow depth constant d= 8 cm (see equations (11) and (12)" => How did the authors calibrate these parameters?
**By comparison with the measured (daily) albedo.**

Page 17, Line 14 (and Page 18, Line 3): "At Exploradores Glacier the measured emissivity reaches values higher than one."=> Please correct LWin before computing the emissivity.
**Ok!**
Page 18, Line 8 (and the end of the paragraph): "The variability of the modeled turbulent fluxes is very similar in all three methods[...] have very similar aspect: in all approaches the sensible heat flux is mainly driven by the temperature difference"=>please refer to Table A1. For instance, if we consider the Bello Glacier, SH is ranging from 6 to 30 W m-2 and LH from -23 to -53 Wm-2. Considering the sum of turbulentheat fluxes, SH+LH is ranging from -40 to +7 W m-2 . This maximum difference between the 3 models (47 W m-2) is larger than those observed in LWnet (22 W m-2)and in SWnet (15 W m-2). Perhaps I did not understand the

end of the paragraph, but the large difference in turbulent heat fluxes results from the very different assumptions done on Ts and on stability corrections.
**Ok! Reformulated!**

Section 5.3: melt rates: I propose that authors include a table to allow a quick validation between modelled ablation and observed ablation from stakes or sonic gauges. In thisTable, it would be interesting to include the elevation of ablation measurements, the latitude, the time period of measurements.
**Thank you for this suggestion. Sadly we do not have ablation data for the modeled summers which is why we have to compare with data from different years and/or different glaciers. This is why we also are not able to judge the results of the different methods on the basis of the observed melt rate. In this sense it would be probably not correct to include a "validation table".**

Page 18, line 23 :" observed melt range"=> "observed melt rates"
**Ok!**
Page 18, line 35: This comparison looks strange because Exploradores is located in campo de Hielo Norte, whereas Grey Glacier is located in Campo de Hielo Sur.
**Again the comparison with the values measured at Grey Glacier are not meant as a "validation" of one special method. We just want to show that the high melt rate that we obtained with one of the methods for Exploradores Glacier is in the range of observed melt rates at Patagonian Glaciers at a similar elevation range. We indicate this more clearly in the manuscript now.**

Page 19, Line 23: "The capacity of the models to reproduce the measured radiative fluxes is still improvable." => This conclusion looks strange when only 3 simple approaches are used. Please note that there is a large number of complex snow models and many complex experiments are done to improve these models (e.g., ESM-SnowMIP (Krinner et al., 2018).
**This statement refers to the two models tested in this contribution and is detailed in the paragraph which follows the statement. Specially the difficulties to transfer model parameterizations for me is a clear indication that even if the energy balance models are trying to reproduce correctly the energy fluxes their parametrizations are based on measurements and they are therefor empirical and not physical . Model Intercomparison Studies like the one you are citing are very valuable but their conclusions are mostly similar to ours: individual model results can be far away from the reality, but, if I use a huge set of models, the average between all these models is mostly doing fine.  But this is due to statistics and not due to good physics!**

Page 19, Line 28 : "This is because the used parameterizations are fits to data that were obtained in different climatic conditions." => This sentence seems trivial. I propose that authors calibrate their model with their observation, and then perform a model sensitivity analysis.

**I am sure that the sentence is not trivial for all the readers of the Cryosphere. Indeed I have seen few publications of measurements of longwave radiative fluxes over glaciers. Again, the goal of this study is not to find THE best parametrization for every glacier (also we indicate them in Figure 10), but to test how transferable or "universal" are the parametrizations proposed in the tested models.**

Page 19 Line 31: "Different parameterizations for the turbulent fluxes of sensible and latent heat were compared in this study,"=> because assumptions made on Ts and equations used were different in the 3 approaches, it is hard to conclude.

**We agree with this statement. This is why we encourage direct measurements of turbulent fluxes over glacier surfaces to be able to better judge the different parameterizations.**

Page 19 Line 33 (and page 20, Line 1): "There are many parameters involved in these parameterizations"=> I don't agree, the bulk method only requires to define z0m.

**In general there exist three different roughness lengths ($z\_0$, $z\_H$, $z\_T$). Formulation was changed!**

Page 20, second paragraph: "Bringing the predicted melt rates by these highly paameterized models in agreement with the observed ones seems to be rather a curve adjustment exercise than a indicator of correct physics." => Perhaps this conclusion is real for EB-model and COSIMA, but I would not extrapolate this conclusion to all the SEB models.

**The SEB models we know, all have a high quantity of non-physical parameters (because their parameterization stem from empirical fits to data).**

Page 20, Line 18: "The inferred melt rates were higher for the Patagonian Andes than for the Central Andes."=> it depends on elevation of study sites. Melt is zero in Patagonia at 4000 m asl.

**Ok, we added: "in the ablation area of the glaciers"**

Page 20, Line 20: "The models underestimated the measured emissivity of the clearsky atmosphere in the Wet Andes."=> please correct LWin before concluding.

**If we firstly "corrected" our model results according to the measurements, then the validation against the measurements would not make sense any more!**

Page 20, line 23 : "To develop or improve physical models we have to validate every single model parameterization against data"=> this could be done in present study.

**We DO it (for the radiative fluxes): for example red dashed line against blue line in Figure 9 or black lines against colored lines in Figure 10.**

---

## Author Response (AR2)

The Paper has been improved and I think the paper could be published after a few more improvements. I am not fully convinced by the authors' responses and I still have some requests about the analysis, and I believe that the manuscript quality could be easily improved:

1. I definitely think that correlations, RMSD or Nash-Sutcliffe Efficiencies between modelled daily values would better support the conclusions of the paper than Figure 5, 6, 7 and 8. Indeed, these figures show a good agreement between models rather than big differences. Even if I partially agree with the fact that "Bringing the melt rates predicted by these highly parameterized models in agreement with the observed ones seems to be rather a curve adjustment exercise than an indicator of correct physics. » I still think that the authors need to present a Table with statistics before reaching this conclusion.

**Ok we added such as table now.**

Figures with hourly values would also help. Perhaps these figures could be included in the appendix (see below).

**Ok we added hourly plot in the appendix now.**

2. I still believe that COSIMA calibration has been done "quickly" on Mocho glacier and is not robust. For instance, the mean modelled SWnet values (from COSIMA model) significantly differ from the measured ones (even on Mocho glacier), whereas the mean values obtained with the very simple assumption of the EB-model (a constant albedo value) correctly fit with mean observations (see Table A1).

**Ok, this point has to clarified: for the Eb-model we used as a constant the measured mean albedo, while for COSIMA we used standard literature values for the ice surfaces and only for Mocho Glacier a calibration of parameters involved in the snow albedo parameterizations was realized. This suggests that model calibration has not been performed in order to allow transferability of the**

parameters. A calibration based on a Monte-Carlo approach and an optimization of scores (e.g., the Nash-Sutcliffe efficiency) using observations from different sites would remove this. This aspect could be improved. I understand that the authors will not make any further full calibration in the present study, but I believe that this is a clear deficiency of the study, because this impedes concluding whether modelling inconsistencies result from the lack of optimization or from model itself.

We agree that the described method could potentially improve the agreement between measured and modeled albedo in Figure9. However it will never be perfect since the measured albedo is not responding to the precipitation dataset as the models proposes it. For example for the second precipitation event (25th to 27th of February), which is rather small in the record, measured albedo increases strongly whilst for the third intense precipitation event (6th and 7th of March) the albedo increases only slightly. This is probably because the precipitation record from the valley is not such a good indicator for the glacier site where the albedo in measured ( we mention that in the text).

As a consequence, I suggest that the authors inform the reader about this lack. After writing that "Openly shared codes are the best way to improve physical models, since everyone can test the individual parametrizations against his data and adjust or improve them accordingly. This is the preferred way to obtain physical parametrizations as opposed to large chains of models which supposedly model physical processes whose individual performance, however, is not validated and the final results rather come out of a black box", I suggest to add that "Moreover, an accurate model optimization is required to allow transferability of the parameters at large scales. This could be done using a Monte Carlo approach (e.g., Mölg et al., 2012) and calculation of scores between measured and modelled surface height changes and/or energy fluxes. Optimization strategies could be defined using field records from the different studied regions. Finally, the parameter transferability in space and time

may be tested using a leave-one-out cross validation approach (e.g., Hofer et al., 2010)."

Here I am not so sure if I can follow you. Are you proposing to statistically look for the best parameter set which minimizes the error of the model in comparison with the measurements? This would guaranty best tranferability for you? In my opinion to guaranty tranferability of the parameters we must physically understand on what these parameters depend and then vary them according to the difference of the speed and magnitude of the physical processes on the different sites.

=> Concerning Authors responses:

>Authors: "What do you mean with "fully" used? To our understanding the bulk aerodynamic approach is a way of quantifying the turbulent fluxes by making three important assumption (which are stated in the manuscript). We are happy to get feedback from you in the case you are thinking that we forget to mention another important assumption of this approach."

My response: My concern was that the authors did not use stability corrections in the reference and in the EB-model. These corrections are crucial to accurately compute the turbulent heat fluxes. In the present version it is still unclear whether the authors include corrections during unstable conditions. What do you mean with unstable conditions? High wind speeds? Then Ri becomes smaller than 0.01 and no correction is applied.

Indeed, Page 11, line 15, the authors write "The same stability correction based on the bulk Richardson number Ri describe in section 3.2 is applied here to account for the reduced vertical exchange of air masses in stable conditions (Braithwaite, 1995b).". Do the authors apply any corrections when Ri < 0?

**Se above: no correction is applied for Ri<0.01**

>Authors: "You are right. This formula was implemented in the first version of COSIMA that we downloaded and was indicated in Huintjes et al. 2015a. But this seemed to be a bug which was changed now."

My response: Please remove the red curve in figure 3 or clearly write in the figure caption that the initial COSIMA relationship was not used and that you used Bolton curve instead.

**Ok. we clarified that in the figure caption now.**

>Authors: "Ok. Since this work is about Chilean Glaciers we restricted the literature review mainly to studies on Chilean Glaciers. We are happy to receive additional recommendations for studies of relevance for our work to include

them in the literature review."

My response: I did not make an exhaustive review, but only for Chile and only during the last 4 years, several papers were published:

Thank you very much for this literature suggestions. With the intention of giving preference to quality over quantity, we try to carefully chose manuscripts which we estimate to be really relevant for our study. Generally we think that studies that only "compute" the energy balance of Andean Glaciers are not automatically relevant for our study. We think that studies that critically analyze the energy balance of glaciers are the ones that are relevant for our study. In this spirit we added some comments on your suggestions below indicating why we think that they are relevant or not.

Weidemann SS, Sauter T, Malz P, Jaña R, Arigony-Neto J, Casassa G and Schneider C (2018) Glacier Mass Changes of Lake-Terminating Grey and Tyndall Glaciers at the Southern Patagonia Icefield Derived From Geodetic Observations and Energy and Mass Balance Modeling. Front. Earth Sci. 6:81. doi: 10.3389/feart.2018.00081

Although the surface energy balance is computed in this contribution (using the COSIMA model), the performed analysis concentrates on mass balance. No critical analysis of the energy fluxes is performed (albedo, emissivity of the atmosphere, surface temperature, or subsurface energy fluxes). Indeed figure 6 is indicating that heat is flowing from the glacier to its surface all year around, which makes not much sense. Here, an older version of the COSIMA model was used which still contained several bugs.

Réveillet, M., MacDonell, S., Gascoin, S., Kinnard, C., Lhermitte, S., and Schaffer, N.: Impact of forcing on sublimation simulations for a high mountain catchment in the semiarid Andes, The Cryosphere, 14, 147–163, https://doi.org/10.5194/tc-14-147-2020, 2020.

**Our simulations show that sublimation is not an important ablation process for the glaciers and time periods treated in our paper.**

AYALA, A., PELLICCIOTTI, F., PELEG, N., & BURLANDO, P. (2017). Melt and surface sublimation across a glacier in a dry environment: Distributed energy-balance modelling of Juncal Norte Glacier, Chile. Journal of Glaciology, 63(241), 803-822. doi:10.1017/jog.2017.46

**Thank you for that suggestion, we think it is a relevant manuscript. We are citing it now.**

Bravo, C., Quincey, D. J., Ross, A. N., Rivera, A., Brock, B., Miles, E., & Silva, A.(2019). Air temperature characteristics, distribution, and impact on modelled ablation for the South Patagonia Icefield. Journal of Geophysical Research: Atmospheres, 124, 907–925. https://doi.org/10.1029/2018JD028857

**This paper is analyzing 10 month of temperature data obtained from 4 stations which are located on a transect of the Southern Patagonian Icefield. We do not see a direct connection to our work, since we are concentrating on the ablation area of the glaciers.**

Bravo, C., Loriaux, T., Rivera, A., and Brock, B. W.: Assessing glacier melt contribution to streamflow at Universidad Glacier, central Andes of Chile, Hydrol. Earth Syst. Sci., 21, 3249–3266, https://doi.org/10.5194/hess-21-3249-2017, 2017.

**We think this a very interesting study, but the focus is on melt and glacier discharge. Energy balance is computed but no critical new insights are given. Also, Universidad Glacier is in a different altitudinal range as compared to the glacier of the Central Andes which we are studying.**

Other studies were published in areas near Chile, but with a clear interest in the present paepr (in particular for turbulent heat fluxes estimations, or for glacier-wide calculations):

Litt, M., Sicart, J., Helgason, W.D. et al. Turbulence Characteristics in the Atmospheric Surface Layer for Different Wind Regimes over the Tropical Zongo Glacier (Bolivia, 16°S). Boundary-Layer Meteorol 154, 471–495 (2015).

**We think that this an interesting and relevant study. We are citing it now. Thank you for the suggestion.**

Maussion, F., Gurgiser, W., Großhauser, M., Kaser, G., and Marzeion, B.: ENSO influence on surface energy and mass balance at Shallap Glacier, Cordillera Blanca, Peru, The Cryosphere, 9, 1663–1683, https://doi.org/10.5194/tc-9-1663-2015, 2015.

**Again we think this is a really interesting study. However it is about the tropical glacier zone of the Andes which is not studied by us. Glacier mass balance dependency on ENSO in this glacier region seems to be inverse to then ENSO dependency of glaciers in the Dry Andes where they are positively correlated (for example Rabatel et al 2011).**

Finally several studies using the SEB at large scale (including Chile), have been published. In these studies, estimates are not clearly validated with field data, but these studies could be discussed (at least Mernild et al. study is frequently cited):

Mernild, S. & Wilson, R. The Andes Cordillera. Part III: glacier surface mass balance and contribution to sea level rise (1979-2014). Int. J. Climatol. 37, 3154–3174 (2016).

**Following our introducing arguments, we think this clearly not a relevant study. Complex model chains are used here but intermediate model outputs are not validates at all. The modeled surface mass balance data are not reproducable and it is by now means clear if the physical processes are modeled well or if the errors of the individual models cancel out nicely.**

L.J. Vargo, J. Galewsky, S. Rupper, D.J. Ward, Sensitivity of glaciation in the arid subtropical Andes to changes in temperature, precipitation, and solar radiation, Global and Planetary Change, Volume 163, 2018, Pages 86-96, https://doi.org/10.1016/j.gloplacha.2018.02.006.

Sagredo, E., Rupper, S., & Lowell, T. (2014). Sensitivities of the equilibrium line altitude to temperature and precipitation changes along the Andes. Quaternary Research, 81(2), 355-366. doi:10.1016/j.yqres.2014.01.008

**We think that both studies are interesting, but they do not directly analyze the energy balance of glaciers.**

>"Page 10 Line 6: please cite : U.S. Army Corps of Engineers (1956).

>Authors: We do not have access to this piece of work!"

My response: The Snow Hydrology book is (at least) available in google books (even if the plate 5.2, presenting albedo variations has not been scanned): https://books.google.fr

**Ok, we prefer to cite literature which is easily accessible for everyone.**

>Authors: "Regarding the great similarity of the course of the melt rates observed in the Figures 6,7,8 we do not think that additional statistics (like correlations and standard deviations) would add some crucial new insights."

My response: I still believe that a table with correlations, RMSE or Nash-Sutcliffe Efficiency values

between daily values of SWnet, albedo, LWnet, SH and LH, between every models would help (in particular, to interpret the modelling of the albedo). Please see other comments below.

**Ok we added a table now which is indicating some statistic.**

**=> "Comparison with previous version:**

>A quick comparison of the results of present and previous versions of the paper demonstrates that assuming a constant albedo value or Tsurf = 0°C (see Table A1) induce very large biases in the modelling results. The use of stability corrections is crucial in the calculation of the turbulent heat fluxes. This demonstrates that results from simple models (as EB-model) are largely erroneous. This could be commented in the text.

I do not fully agree here: the fact of using constant albedo is not automatically introducing very large biases. SW\_net is reproduced very well by the EB-Model, where the mean albedo is chosen as constant. Regarding Tsurf =  $0^{\circ}$ C: biases are up to  $21W/m^2$ , which is important. We discuss this in section 5.2 (second last paragraph). EB-Model is also applying stability corrections, since the transfer coefficients depend on the Monin-Obukhov length scale (see section 3.3 formulas 7 and 8).

>In the models (e.g. in the reference model), I understand that melt occurs when Tsurf = 0°C and SEB = R + SH + LH > 0. However, how do the authors consider the refreezing and the frigories stored during the night or when the SEB is negative or very close from 0 (at the end of the melt season (see figure 6,7,8)? Melt can occur at the surface when Tsurf = 0°C, while the subsurface temperature is still below 0°C. But, in that case, the surface melting is reduced because incoming shortwave radiation penetrates in the ice and heat condition warms the snow/ice layers below the surface. This induces that Melting amount is less than R + SH + LH until frigories are removed from the subsurface layers. How is this process considered in the simple reference and EB-models? Please comment this point in the text.

**In the reference database and EB-Model negative energy balance during night times are considered for the computation of the daily melt rates.**

=> Minor remarks:

>In the abstract, the authors write that "The influence of the stability correction and the roughness length on the magnitude of the turbulent fluxes in the different climate settings was examined." => This conclusion is almost not developed in the text and relies on Table A1, which is found in the appendix. Please move Table A1 in the main text and develop the discussion on this point.

**Ok we moved the table into the main text now. The influence of the stability correction and the roughness length on the magnitude of the turbulent fluxes in the different climate settings is discussed in section 5.2 (last paragraph)**

>Page 2, Line 35 : Sensibel => sensible Please remove other typos

**ok**

>Page 4 Last line : "The projections of future changes in climate depend on the different climatological/glaciological zones. This is why a detailed analysis of the processes that determine the energy exchange at the surface of the glaciers in the different climatological zones is necessary, to be

able to make reliable predictions of future surface mass balance and melt water discharge of Chilean glaciers." => please, move this sentence at the beginning of this section.

**ok**

>Page 6, line 15: "Hourly data were generated using a matlab interpolation scheme." => What is "a matlab interpolation scheme" ? what kind of scheme did you use?

**Ok. we are more precise now: ... using the linear interp1 matlab interpolation scheme.**

>Page 7 line 2: "incoming and outgoing longwave radiative fluxes (table 3)" => Table 3 refers to surface roughness lengths.

**Ok. There were different wrong references due to the new table 3. This was corrected now.**

>Page 8, line 6 :" Assumption 1. is normally fullfilled for a neutral atmosphere, but, over a glacier surface, the temperature gradient is often inverted (especially during summer). This stable layering of air masses reduces the vertical exchange specially for low wind speeds" => what do the authors mean with an inverted gradient? The surface is warmer than above? Do they mean unstable conditions? If it were the case, there would be a contradiction in the sentence. Please explain and refer directly to stable or unstable conditions. Moreover, did they apply corrections when Ri<0?

**Ok now we explain better the concepts of a neutral atmosphere and an inverted temperature gradient.**

>Page 13, line 5 : Figure Cs4?

**Figure 4**

>Page 13, Line 11: radiation show => radiation shows

**ok**

>Page 14, line 15: "The predicted melt rates for the specific study points (locations of the AWS) in the ablation area of the glaciers are higher for the Patagonian Glaciers as compared to the glaciers of the Central Andes" and Page 22, line 1: "The inferred melt rates in the ablation area of the glaciers were higher for the Patagonian Andes than for the Central Andes." => as written in my first review, I don't agree with this sentence: it depends on elevation. The only way to conclude would be to plot the mean melting vs. mean temperature, or Melting vs. elevation difference between the AWS and the ELA. Actually, if the authors display this figure, they will observe that there is no possible comparison between sites: for instance, melting at the Bello Glacier was similar to the Tyndall Glacier when temperature was 5°C lower at The Bello Glacier. This figure could be presented in the appendix.

Thank you for this suggestion. Below the two suggested figures + a plot of melt rates against relative elevation difference from ELA (ELA-z\_station)/(elevation range of the glacier)

We can observe several interesting in these figures:

a) We can see that in the Wet Andes melt rates clearly depend on the temperature, while for the Central Andes this trend is not visible.

b) As expected: in the Wet Andes melt rates increase with elevation difference from ELA, however this is not the case for the two glaciers of the Dry Andes. At similar elevation difference from ELA melt rates are slightly higher in the Wet Andes.

c) If we modify the graph b) slightly and plot melt rates against relative elevation difference from ELA in both zones melt rates increase with relative elevation difference. However melt rates in the Wet Andes are generally higher.

We included plots a) and c) in section 5.4 now and changed the formulation of the conclusion accordingly.

Section 5.2 : Parametrizations of the surface energy fluxes : Again, I propose to include table A1 directly in the main text.

**Ok, table was moved to the main text**

I also propose to include a table with statistics in the appendix, and to discuss this new table in the text. **Ok, a table with statistic was included.**

These tables are important to understand the discussion. For instance: Section 5.5 page 21, line 2: "The albedo aging effect implemented in COSIMA is a big improvement regarding to constant albedo parametrizations for snow, firn and ice surfaces. » => after analysing SWnet values in Table A1, this sentence looks strange. It seems that a constant albedo gives better results that a varying albedo.

**As explained above, for EB-Model the mean measured albedo was used, while for the COSIMA model standard values for ice albedo were used. We are not referring to the table with this sentence.**

A quick statistical analysis, using a RMSD or a Nash–Sutcliffe efficiency calculations done on SWnet values would help to characterize whether the albedo model improves the quality of the modelling (i.e. using the albedo model gives better results than assuming a constant albedo value).

Actually, I already asked information on albedo calibration (and more generally on calibration of the model parameters) in my first review, but it seems that my question was not clear.

>The authors answered:

>"By comparison with the measured (daily) albedo."

How did the authors obtain the parameters given in the caption of Figure 9? Did they perform a monte carlo approach with data from Mocho glacier? Or did they realize a quick estimation of parameters allowing the mean modelled albedo to approximately equal the mean measured values at Mocho?

**We calibrated the parameters using the following procedure**

- 1. chose alpha\_frsnow in a way that it agrees well with albedo maxima observed after snowfall
- 2. chose alpha\_firn in a way that it agrees with the minima of the peaks
- 3. Chose t\* in such a way that the it fits the observed albedo decrease after snowfall
- 4. Chose d\* in way that the albedo increase in triggered correctly by the precipitation.

Finally, please give references to discuss the values given in the caption of Figure 9 (These values are likely representative for an area and type of glacier elsewhere).

**Ok. We are indicating now how our values compare to values chosen in other studies (Moelg et al 2012, Huitjes et a. 2015).**

>Section 5.5, Page 21, Line 1: "The capacity of the models to reproduce the measured radiative fluxes is still improvable." => The authors never compare the fluxes at an hourly timescale, but such a figure would clearly help to see model discrepancies. In particular, if the measured hourly (uncorrected and corrected) values of LWout were displayed and compared with those from the COSIMA model, it would help to see whether the "constant " correction on LWout (and LWin) data is accurate or not. A comparison over a 5-day time period for one (or every glaciers) could be presented in the appendix.

**Ok. Hourly comparison were added now.**

>Page 19, line 3 : paramtrizations => parametrizations

**Ok**

>Page 20, line6 : CR4 => CNR4 ?

Ok

>Page 20, Line 8 : this in in very good agreement => this is in good agreement

**Ok**

>Page 20, line 30 : higer => higher

Ok

>Page 20, lines 10-14 : I don't understand the comparison between the Grey glacier and the Exploradores glacier. Elevation of the sites is very different and the glaciers are from two different icefields. Please remove this and use data from NPI instead. Why did the authors use these data rather than those used in Figure 11 from Schaefer, et al., (2013)?

The measurements at Grey Glacier are interesting, because the measurement period is from January to March, that is the summer period in which this study is focusing. We mention the ablation measurements at Nef and Moreno glaciers (shown in Figure 11 Schaefer, et al., (2013)) now.

>Line 1 page 18 : "A drawback of this comparison is certainly that we do not know the exact amount of snow falling on the glacier, but deduce it from the liquid precipitation measured at a automatic weather station in the valley." => I propose to include that "Moreover, a more detailed modelling of albedo accounting for snow metamorphism could be tested to see whether it reproduces more accurately the short timescale variations".

We agree generally with this sentence, however a necessary condition for this detailed study are detailed measurements of the precipitation falling in situ including snow height, snow water equivalent, grain sizes, ...

**=> References:**

Hofer, M., Mölg, T., Marzeion, B., and Kaser, G.: Empirical statistical downscaling of reanalysis data to high resolution air temperature and specific humidity above a glacier surface (Cordillera Blanca, Peru), J. Geophys. Res., 115, D12120, doi:10.1029/2009JD012556, 2010.

Mölg, T., Maussion, F., Yang, W., and Scherer, D.: The footprint of Asian monsoon dynamics in the mass and energy balance of a Tibetan glacier, The Cryosphere, 6, 1445–1461, doi:10.5194/tc-6-1445-2012, 2012.

Schaefer, M., H. Machguth, M. Falvey, and G. Casassa (2013), Modeling past and future surface mass balance of the Northern Patagonia Icefield, J. Geophys. Res. Earth Surf., 118, 571–588, doi:10.1002/jgrf.20038.

**Ok, sentence was reformulated and shortened.**

The most relevant changes in the manuscript during this second revision were:

- adding Figure11 where we show the dependency of the modeled melt rates on different para meters
- adding Figure A2, in which we show the hourly variation of the modeled energy fluxes
- adding Table A1, where we present the statistics of the comparison of the different methods to compute the energy fluxes and melt rates.
- Adding some conclusions about the dependency of the modeled melt rates on air temperature and relative elevation difference with ELA in the different glaciological zones.

[revised manuscript text omitted]